# Cross-Scale Self-Supervised Blind Image Deblurring via Implicit Neural Representation

**Tianjing Zhang[1], Yuhui Quan[2], Hui Ji[1]**
[1] Department of Mathematics, National University of Singapore
[2] School of Computer Science and Engineering, South China University of Technology
`tianjingzhang@u.nus.edu, csyhquan@scut.edu.cn, matjh@nus.edu.sg`

## Abstract

Blind image deblurring (BID) is an important yet challenging image recovery problem. Most existing deep learning methods require supervised training with ground truth (GT) images. This paper introduces a self-supervised method for BID that does not require GT images. The key challenge is to regularize the training to prevent over-fitting due to the absence of GT images. By leveraging an exact relationship among the blurred image, latent image, and blur kernel across consecutive scales, we propose an effective cross-scale consistency loss. This is implemented by representing the image and kernel with implicit neural representations (INRs), whose resolution-free property enables consistent yet efficient computation for network training across multiple scales. Combined with a progressively coarse-to-fine training scheme, the proposed method significantly outperforms existing self-supervised methods in extensive experiments.

## 1 Introduction

Uniform blurring, a degradation commonly encountered in optics, leads to the loss of important details within a captured image. Uniform blurring usually can be described as the convolution:

$$\boldsymbol{y} = \boldsymbol{k} \otimes \boldsymbol{x} + \boldsymbol{n}, \tag{1}$$

where $\otimes$ denotes the 2D discrete convolution operation, $(\boldsymbol{y}, \boldsymbol{x})$ is the pair of blurred and sharp (latent) images. $\boldsymbol{k}$ represents the blur kernel responsible for degradation, and $\boldsymbol{n}$ denotes measurement noise. For example, in digital photography, motion blur is caused by camera shake during exposure. When scene depth variation is small and camera movement is mainly translational within the image plane, motion blur can be approximated by the convolution model (1), with $\boldsymbol{k}$ representing camera motion. In fluorescence microscopy, specimens stained with fluorescent dyes and exposed to specific wavelength light often exhibit blurring due to light diffraction and optical path imperfections, such as optical component misalignment or lens aberrations. These factors obscure critical details, like micro-tubule arrangements in cells. Such blurring can also be modeled by (1). In either case, both the latent image $\boldsymbol{x}$ and blur kernel $\boldsymbol{k}$ are unknown and must be estimated from the blurred image $\boldsymbol{y}$.

BID aims at estimating $(\boldsymbol{k}, \boldsymbol{x})$, the pair of latent image and blur kernel, from the degraded image $\boldsymbol{y}$. BID is a challenging non-linear inverse problem with many plausible solutions due to its inherent solution ambiguity. This ambiguity stems from the fact that the kernel can be decomposed into $\boldsymbol{k} = \boldsymbol{k}_1 \otimes \boldsymbol{k}_2$, suggesting that the pair $(\boldsymbol{k}_1, \boldsymbol{k}_2 \otimes \boldsymbol{x})$ are also viable solutions since $\boldsymbol{y} = (\boldsymbol{k}_1 \otimes \boldsymbol{k}_2) \otimes \boldsymbol{x} = \boldsymbol{k}_1 \otimes (\boldsymbol{k}_2 \otimes \boldsymbol{x})$. One such example is the pair $(\boldsymbol{\delta}, \boldsymbol{y} = \boldsymbol{k}, \boldsymbol{x})$, where $\boldsymbol{\delta}$ denotes the delta kernel, which gives a trivial solution saying a blurred image is the convolution of a blurred image and a delta kernel. This ambiguity indicates the ill-posed-ness of the BID problem.

### 1.1 Discussion on existing approaches for BID

The existing deep learning methods for BID can be roughly classified into the following categories:

38th Conference on Neural Information Processing Systems (NeurIPS 2024).

- *Supervised deep learning for BID with paired data*: Supervised learning methods (*e.g.*, [50, 78, 68, 34, 69]) trains a neural network (NN) using paired data of blurred images (or blur kernels) and GT images, to predict the latent image or/and blur kernel from an input blurred image. These methods mainly depend on a large amount of paired data for training, limiting their application to specific scenarios where such paired data is challenging to collect.

- *Supervised deep learning for BID with un-paired data*: Some works (*e.g.* [36, 61]) use generative adversarial networks (GANs) to train on unpaired blurred and latent images. Although effective for domain-specific images like faces and text, GANs perform poorly on general natural images due to domain shift. Additionally, GAN-based methods often suffer from mode collapse and training instability, resulting in suboptimal outcomes.

- *BID with pre-trained generative model*: These methods employ pre-trained generative models, such as diffusion models [13], which are well-suited for processing images with structural consistency features like human faces. However, their reliance on pre-trained models to generate outputs that conform to the learned image distribution can lead to inauthentic results. This is particularly problematic in fields that require high fidelity, such as medical imaging and microscopy.

- *Self-supervised learning for BID*: To circumvent the challenges of data collection and mitigate potential biases and inauthentic outcomes from generative models, a growing body of research (*e.g.*, [1, 47, 32, 33, 9, 18, 77]) focuses on developing self-supervised deep learning approaches for BID that do not require training dataset or pre-trained model.

In this paper, we focus on self-supervised BID, which is challenging due to the lack of GT images but offers many practical benefits. Most existing works on self-supervised BID are based on the *deep image prior* (DIP) [57] of convolutional NNs (CNNs), which introduces the implicit prior from CNN for preferring structured patterns over random noise during training. The DIP-based self-supervised BID methods typically use two NN-based generators, $\mathcal{G}_{\boldsymbol{k}}$ and $\mathcal{G}_{\boldsymbol{x}}$, to re-parameterize the blur kernel $\boldsymbol{k}$ and the latent sharp image $\boldsymbol{x}$. The generators are then trained to maximize the likelihood of the blurred image $\boldsymbol{y}$, minimizing the following self-supervised reconstruction loss:

$$\mathcal{L}_{sr}(\Theta_{\boldsymbol{k}}, \Theta_{\boldsymbol{x}}) := ||\mathcal{G}_{\boldsymbol{k}}(\cdot; \Theta_{\boldsymbol{k}}) \otimes \mathcal{G}_{\boldsymbol{x}}(\cdot; \Theta_{\boldsymbol{x}}) - \boldsymbol{y}||_2^2. \tag{2}$$

For instance, SelfDeblur [47] employed a CNN for $\mathcal{G}_{\boldsymbol{x}}$ and a Multi-Layer Perceptron (MLP) for $\mathcal{G}_{\boldsymbol{k}}$. MCEM [33] used U-Net for both $\mathcal{G}_{\boldsymbol{x}}$ and $\mathcal{G}_{\boldsymbol{k}}$, and Zhuang *et al.* [77] used INR for both $\mathcal{G}_{\boldsymbol{x}}$ and $\mathcal{G}_{\boldsymbol{k}}$.

Despite lacking access to GT images, these self-supervised BID methods perform competitively against supervised or pre-trained model based approaches, particularly with severely blurred images. However, their performance is less impressive on modestly blurred images and on real-world images. There is practical need for further studies to improve the performance of self-supervised BID across various blurring degrees and real-world images from different optics systems.

## 1.2 Main Idea and Contributions

Without GT images, self-supervised BID methods must address two vital questions for NN training:

1. Without accessing GT images, how to formulate a self-supervised loss to teach the NN-based generators to accurately predict the latent image and kernel from only the blurred image?

2. The non-linear structure of BID makes training NN-based generators challenging. How can we efficiently train them to ensure accurate convergence to the latent images and kernels?

Our answer to Question 1 is a cross-scale loss function that leverages the cross-scale consistency of the estimates from consecutive scales for regularization. Our answer to Question 2 is a progressive cross-scale training scheme to training the NNs, enhancing training efficiency and ensuring the convergence to GT image/kernel.

**Resolution-free INR for effective cross-scale interaction in BID:** Our proposed method is built on re-parametrizatrion of both latent images and kernels by INR [53, 51]. INR represents signals as continuous functions rather than discrete valuel arrays, which in our case is expressed as follows,

$$\boldsymbol{k}(i, j) = \Phi_{\boldsymbol{k}}(i, j; \Theta_{\boldsymbol{k}}) \quad \text{and} \quad \boldsymbol{x}(i, j) = \Phi_{\boldsymbol{x}}(i, j; \Theta_{\boldsymbol{x}}), \tag{3}$$

where $(i, j) \in \mathbb{R}^2$ denotes the spatial co-ordinates of images/kernels, and $\Theta_{\boldsymbol{k}}, \Theta_{\boldsymbol{x}}$ denotes the weights of NNs $\Phi_{\boldsymbol{k}}, \Phi_{\boldsymbol{x}}$ for the kernel and image, respectively.

Using INR for representing images/kernels is due to its inherent resolution-free property. By enabling the model to generate the prediction with higher/lower resolutions from the same learned model, INR facilitates seamless multi-scale processing and cross-scale interaction. In contrast, DIP-based NNs that directly maps noise to an image handle cross-scale interaction clumsily, requiring manual re-scaling or interpolation, which likely introduces artifacts in the prediction.

**Self-supervised cross-scale loss for BID:** Without GT images, the sole constraint we have for training is $\boldsymbol{y} = \boldsymbol{k} \otimes \boldsymbol{x}$. Additional priors for regularization are necessary to alleviate over-fitting. Historically, the down-sampled version of $\boldsymbol{y}$, denoted as $\boldsymbol{y}_{\downarrow_s}$ for scale $s$, has often been used to initiate the blur kernel estimate. However, it is important to note that the blurred image after down-sampling cannot be modeled by a convolution between the down-sampled image and a blur kernel:

$$(\boldsymbol{x} \otimes \boldsymbol{k})\downarrow_2 \neq \boldsymbol{x}\downarrow_2 \otimes \boldsymbol{k}\downarrow_2.$$

In this paper, we present a cross-scale constraint that accurately characterizes the connection between $(\boldsymbol{y}, \boldsymbol{x}, \boldsymbol{k})$ at different scales. For instance, when $s = 2$, the cross-scale constraint is

$$(\boldsymbol{x}\downarrow_2) \otimes (\boldsymbol{k}\downarrow_2) = \frac{1}{4}\big((\boldsymbol{x} \otimes \boldsymbol{k})\downarrow 2) + \sum_{d=1}^{3}(\boldsymbol{x} \otimes \boldsymbol{g}_d)\downarrow_2\big) = \frac{1}{4}\big(\boldsymbol{y}\downarrow_2 + \sum_{d=1}^{3}(\boldsymbol{x} \otimes \boldsymbol{g}_d)\downarrow_2\big),$$

where $\{\boldsymbol{g}_d\}_{d=1}^{3}$ denote three quadrature mirror filters (QMFs) [59] of the kernel $\boldsymbol{k}$. This constraint can be easily implemented with INR-based resolution-free generators, and helps regularize the training of two generators to prevent likely over-fitting caused by the lack of GT images in loss function.

**Progressive cross-scale learning for BID**: A well-established practice in traditional alternating iterative methods of BID to avoid convergence to trivial solutions involves solving the problem in a coarse-to-fine manner [67, 71]. This means the kernel is initially estimated for the blurred image at a coarse scale, followed by propagation of the kernel estimate to finer scales. This approach has proven effective in preventing iterations from converging to trivial solutions.

Despite its effectiveness, the coarse-to-fine strategy remains under-exploited in existing self-supervised BID methods, likely due to the resolution-fixed limitations of CNN or MLP based re-parametrization. Utilizing INR's resolution-free properties, we introduce a progressive learning strategy that begins training INR-based generators at a coarser scale and then refines training at finer scales. This scheme effectively addresses over-fitting and ensures convergence to the truth.

**Main contribution:** Our main contributions are summarized as follows:

- Leveraging the resolution-independent properties of INR for latent images/kernels, we propose a self-supervised cross-scale loss for training the NN without requiring GT images.
- We introduce a progressive multi-scale learning approach for BID, specifically designed to mitigate potential over-fitting due to the non-linear nature of the BID problem.

Extensive experiments conducted on a variety of datasets reveals that our proposed method outperforms existing self-supervised BID techniques as well as the supervised alternatives.

## 2  Related Works

**Traditional non-learning methods for BID:** Before deep learning became prevalent, regularization methods were the prominent approach for BID. These methods resolve solution ambiguities by imposing pre-defined priors on images and blur kernels, such as image gradient sparsity [8, 4, 24, 5, 67], image patch recurrence [54, 38], and Laplace priors in dark channels [42, 70]. These methods are based on some iteration scheme, and edge selection is an effective technique for better robustness and stability of the iteration [12, 65, 41, 15, 71]. Since regularization methods can be recast as Maximum-A-Posteriori (MAP) estimators in Bayesian inference, another class of BID methods is derived from variational Bayesian estimators [35, 14, 28, 63, 2], as well as variational expectation maximization [29, 71]. The success of these methods requires rigorous tuning of hyper-parameters related to priors. In contrast, deep learning based methods can automatically learn priors from data.

**Supervised deep learning for BID:** In recent years, there has been rapid progress in supervised deep learning methods for BID. By training over many pairs of blurred/truth images, these methods either explicitly estimate the blur kernel (*e.g.*, [50, 7, 40, 34] or only estimate latent images (*e.g.*,

[66, 56, 25, 75, 20, 6, 11, 73, 72]). The former is more efficient for handling uniform blurring. The latter is more general and can handle non-uniform blurring.

**Deep learning for BID with unpaired data or pre-trained model:** There are also methods that are trained on unpaired dataset. Lu *et al.* [36] train the GAN for domain-specific deblurring. Wen *et al.* [61] propose a structure-aware deblurring method, and Chung *et al.* utilize generative priors from diffusion models to jointly estimate the blur kernel and latent images [13]. Nevertheless, these methods still require GT images in unpaired data for training or depend on pre-trained models, which can be challenging and present difficulties in adaptation. In contrast, our method goes through a self-supervised manner, addressing this limitation.

**Self-supervised deep learning for BID:** These methods address the issues of data collection and dataset bias in supervised methods. Built on the DIP prior for image/kernel, Ren *et al.* [47] introduced SelfDeblur, leveraging two NN-based generators trained through a loss with optional TV regularization. The ensemble NN [9] aggregates deblurring outcomes from multiple NNs to improve performance. Li *et al.* [32] proposed using Monte-Carlo methods to sample NN weights as an approximation of the MAP estimator of images and kernels. Li *et al.* [33] presented a self-supervised training scheme derived from the EM method. Dong *et al.* [18] combined the implicit prior from NN architecture and hand-crafted prior to regularize the NN training. Zhuang *et al.* [77] re-parametrized images and kernels by INR, and reply on the implicit prior induced by INR and early stopping for regularization. Built on resolution-free INR, we propose a cross-scale self-supervised loss function and an efficient coarse-to-fine training scheme for self-supervised BID.

Most self-supervised BID methods are limited to handling uniform blurring, with Li *et al.* [33] being the only one capable of addressing both uniform and non-uniform blurring caused camera shake. Additionally, self-supervised super-resolution methods[3, 55] also involve the estimation of blur kernel. While both estimate the blur kernel. However, these methods differ in their input: low-resolution images versus high-resolution images. Moreover, the blur degree in BID usually is much more severe than in super-resolution.

**Coarse-to-fine estimation for BID:** Coarse-to-fine schemes, which gradually refine the at different scales, have proven effective in traditional alternating iterative methods [65, 12, 52]. The traditional multi-scale methods typically use estimates from coarser scales solely as initial estimates for finer scales. In contrast, our approach for self-supervised deep BID leverages the cross-scale interaction of estimates with exact relations, rather than approximations. Coarse-to-fine scheme used in recent supervised NN-based methods for BID (*e.g.,* [39, 56, 74, 11, 73]) train the NN such that the estimates fit well the input image at different scales. Our multi-scale scheme is more for introducing a new scale consistency loss function specifically tailored for self-supervised BID. This allows for more precise and consistent refinement of the estimations across scales.

**INR for image recovery:** In the domain of images, INR [10, 37, 43] encodes images as NN weights, mapping coordinates to pixel values for a compact, continuous representation. There are also multi-scale extensions [27, 49] of INR for more efficient image compression. It has been used for image restoration tasks such as image in-painting, denoising, and super-resolution [51, 21, 64]. Besides images, INR also has been used for encoding defocus blur kernels [46] and motion blur kernels [77]. Our work is the first to exploit the resolution-free properties of INR for BID, leveraging these properties for a multi-scale approach and cross-scale interaction.

## 3   Methodology

In this paper, we propose an INR-based progressive cross-scale self-supervised BID method. We first introduce the INR-based modeling for latent images and blur kernels, followed by the formulation of the self-supervised cross-scale loss function and the progressive learning strategy.

**Double-INR model for BID:** In our approach, the blur kernel $k$ and the latent image $x$ are re-parameterized by two INR models, $\Phi_k$ and $\Phi_x$, respectively. Each model maps a spatial coordinate $[i, j]$ to a pixel value. Let $\mathbb{I}_k, \mathbb{I}_x \subset \mathbb{Z}^2$ denote the sets of spatial coordinates within the feasible domain for the blur kernel $k$ and the latent image $x$, respectively. Then, they can be expressed as

$$\begin{cases} k[\mathbb{I}_k] = \Phi_k(\mathbb{I}_k; \Theta_k) & : \quad k[i, j] = \Phi_k\big([i, j]\big), \, [i, j] \in \mathbb{I}_k; \\ x[\mathbb{I}_x] = \Phi_x(\mathbb{I}_x; \Theta_x) & : \quad x[i, j] = \Phi_x\big([i, j]\big), \, [i, j] \in \mathbb{I}_x, \end{cases} \tag{4}$$

where $\Theta_{\boldsymbol{k}}, \Theta_{\boldsymbol{x}}$ denote the NN weights of $\boldsymbol{k}, \boldsymbol{x}$. The INR-based re-parameterization (4) allows for the generation of kernel and image at any coordinates, providing a representation at arbitrary scales.

**INR-based multi-scale representation:** Let $\downarrow_m$ denotes the standard down-sampling operator:

$$(\boldsymbol{k}\downarrow_m)[i,j] = \boldsymbol{k}[i \cdot m, j \cdot m] \quad \text{and} \quad (\boldsymbol{x}\downarrow_m)[i,j] = \boldsymbol{x}[i \cdot m, j \cdot m].$$

Then, we can form both the kernel and the image in a dyadic pyramid, from the original scale and to coarser scales: Let $\boldsymbol{k}^{(0)} = \boldsymbol{k}, \boldsymbol{x}^{(0)} = \boldsymbol{x}$. Then define

$$\boldsymbol{k}^{(s)} = (\boldsymbol{k}^{(s-1)}) \downarrow_2 \quad \text{and} \quad \boldsymbol{x}^{(s)} = (\boldsymbol{x}^{(s-1)}) \downarrow_2, \quad \text{for } 1 \leq s \leq S_0. \tag{5}$$

For any co-ordinate set $\mathbb{I}$, we define its co-ordinate set at scale $s$ as $\mathbb{I}^{(s)} = \{[i,j] : [2^s i, 2^s j] \in \mathbb{I}\}$. Then, using INR-based model (4), $\boldsymbol{k}^{(s)}$ and $\boldsymbol{x}^{(s)}$ at scale $s$ can be expressed as

$$\boldsymbol{k}^{(s)}[\mathbb{I}_{\boldsymbol{k}}^{(s)}] = \boldsymbol{k}[2^s \mathbb{I}_{\boldsymbol{k}}^{(s)}] = \Phi_{\boldsymbol{k}}\big(2^s \mathbb{I}_{\boldsymbol{k}}; \Theta_{\boldsymbol{k}}\big) \quad \text{and} \quad \boldsymbol{x}^{(s)}[\mathbb{I}_{\boldsymbol{x}}^{(s)}] = \boldsymbol{x}[2^s \mathbb{I}_{\boldsymbol{x}}^{(s)}] = \Phi_{\boldsymbol{x}}\big(2^s \mathbb{I}_{\boldsymbol{x}}; \Theta_{\boldsymbol{x}}\big). \tag{6}$$

Such a multi-scale representation can facilitate the training of the INR-based generators for BID, as it allows for the generation of image details at arbitrary scales.

## 3.1 NN architecture of INR-based generators for kernel/image

**Kernel generator:** The kernel generator $\Phi_{\boldsymbol{k}}$ in our approach is a three-layer MLP with 128 feature nodes that takes coordinates normalized to $[-1, 1]$ as input and adopts sinusoidal activation functions in every layer, following the Sinusoidal Representation Networks (SIRENs) [53]. The output layer employs a Softmax activation function to ensure the prediction satisfies the two physical constraints:

$$\text{Non-negativity: } \boldsymbol{k}[i,j] \geq 0 \text{ for all } [i,j]; \quad \text{Normalization: } \sum_{i,j} \boldsymbol{k}[i,j] = 1. \tag{7}$$

Then, a Kernel Centering layer is applied to address possible positional shifts in estimated kernels for better training stability. For a kernel $\widetilde{\boldsymbol{k}} \in \mathbb{R}^{H \times W}$ satisfying (7), calculating its centroids $(\boldsymbol{c}_x, \boldsymbol{c}_y)$ by

$$\boldsymbol{c}_y = \sum_{i,j} i \cdot \widetilde{\boldsymbol{k}}(i,j), \quad \boldsymbol{c}_x = \sum_{i,j} j \cdot \widetilde{\boldsymbol{k}}(i,j)$$

The kernel $\widetilde{\boldsymbol{k}}$ is then shifted by $[\lfloor H/2 \rfloor - \boldsymbol{c}_y, \lfloor W/2 \rfloor - \boldsymbol{c}_x]$ to ensure it is centered geometrically.

**Image generator:** The image generator $\Phi_{\boldsymbol{x}}$ adopts a U-Net structure with five blocks, separated by down- or up-sampling layers and connected by skip connections. Each block integrates a sequence of Convolution, Batch Normalization, and ReLU. The NN concludes with a $1 \times 1$ convolution layer followed by a Sigmoid layer to ensure the output image values remain within $[0, 1]$. To efficiently generate high-frequencies of images, following [51], the input spatial coordinates are first transformed into a higher dimensional space using a high-frequency function $\gamma(\cdot)$ (sinusoidal):

$$\gamma(\boldsymbol{p}) = \big(\boldsymbol{p}, \sin\big(2^0 \pi \boldsymbol{p}\big), \cdots \sin\big(2^{L-1} \pi \boldsymbol{p}\big), \cos\big(2^{L-1} \pi \boldsymbol{p}\big)\big), \tag{8}$$

where $\boldsymbol{p} := (i, j)$ represents the normalized coordinate values within $[-1, 1]$, and $L$ is an positive integer. Note this encoding operation is used only in $\Phi_{\boldsymbol{x}}$, not in $\Phi_{\boldsymbol{k}}$.

## 3.2 Self-Supervised scale consistency loss

Without GT images, the only readily available loss function to train the generators is the fitting loss:

$$L_{\text{fit}}(\Theta_{\boldsymbol{k}}, \Theta_{\boldsymbol{x}}) = \mathcal{M}_f\big(\boldsymbol{y} - \boldsymbol{k} \otimes \boldsymbol{x}\big) = \mathcal{M}_{\text{fit}}\Big(\Phi_{\boldsymbol{k}}(\mathbb{I}_{\boldsymbol{k}}; \Theta_{\boldsymbol{k}}) \otimes \Phi_{\boldsymbol{x}}(\mathbb{I}_{\boldsymbol{x}}; \Theta_{\boldsymbol{x}}), \boldsymbol{y}\Big), \tag{9}$$

where $\mathcal{M}_f(\cdot)$ is some distance metric. Such a fitting loss clearly is not sufficient to resolve solution ambiguities in BID. To address this, we introduce a scale consistency loss that regularize the training by enforcing cross-scale consistency of the estimation, which is based on the following proposition.

**Proposition 1.** *For a kernel (filter) $\boldsymbol{k}$, let $\boldsymbol{g}_1, \boldsymbol{g}_2, \boldsymbol{g}_3$ denote its associated QMF filters [59] defined by*

$$\boldsymbol{g}_1[m,n] = (-1)^m \boldsymbol{k}[m,n], \ \boldsymbol{g}_2[m,n] = (-1)^n \boldsymbol{k}[m,n], \ \boldsymbol{g}_3[m,n] = (-1)^{m+n} \boldsymbol{k}[m,n], \tag{10}$$

*for any $[m,n] \in \mathbb{I}_{\boldsymbol{k}}$. Then, we have the following relation between consecutive two dyadic scales:*

$$(\boldsymbol{x}\downarrow_2) \otimes (\boldsymbol{k}\downarrow_2) = \frac{1}{4}\big((\boldsymbol{x} \otimes \boldsymbol{k})\downarrow_2 + \sum_{d=1}^{3} (\boldsymbol{x} \otimes \boldsymbol{g}_d)\downarrow_2\big). \tag{11}$$

*Proof.* See Appendix A for the proof. □

As seen from Proposition 1, the down-sampled blurred image $\boldsymbol{y}\!\downarrow_2$ does not equal to the convolution of the down-sampled latent image $\boldsymbol{x}\!\downarrow_2$ and the down-sampled kernel $\boldsymbol{k}\!\downarrow_2$, incurring additional term $\sum_{d=1}^{3}(\boldsymbol{x}\otimes\boldsymbol{g}_d)\!\downarrow_2$. Therefore, down-sampled blurred images are fine for some initial estimation, not for regularizing the NN to obtain accurate estimation. To address this, based on Proposition 1, we introduce a scale consistency loss across two consecutive scales: for each scale $s$,

$$L_{\mathrm{cross}}^{(s)}(\Theta_{\boldsymbol{k}},\Theta_{\boldsymbol{x}}) = \mathcal{M}_c\Big(4(\boldsymbol{x}^{(s)}\!\downarrow_2)\otimes(\boldsymbol{k}^{(s)}\!\downarrow_2),(\boldsymbol{x}^{(s)}\otimes\boldsymbol{k}^{(s)})\!\downarrow_2 + \sum_{1\leq d\leq 3}(\boldsymbol{x}^{(s)}\otimes\boldsymbol{g}_d^{(s)})\!\downarrow_2\Big) \quad (12)$$

$$= \mathcal{M}_c\Big(4(\boldsymbol{x}^{(s+1)})\otimes(\boldsymbol{k}^{(s+1)}),(\boldsymbol{x}^{(s)}\otimes\boldsymbol{k}^{(s)})\!\downarrow_2 + \sum_{1\leq d\leq 3}(\boldsymbol{x}^{(s)}\otimes\boldsymbol{g}_d^{(s)})\!\downarrow_2\Big), \quad (13)$$

where $\{\boldsymbol{g}_d^{(s)}\}_{d=1}^{3}$ denotes the QMF filter bank of the kernel $\boldsymbol{k}^{(s)}$ defined by (10), and by (6),

$$\boldsymbol{k}^{(s)}[\mathbb{I}_{\boldsymbol{k}}^{(s)}] = \Phi_{\boldsymbol{k}}\big(2^s\mathbb{I}_{\boldsymbol{k}}^{(s)};\Theta_{\boldsymbol{k}}\big) \quad\text{and}\quad \boldsymbol{x}^{(s)}[\mathbb{I}_{\boldsymbol{x}}^{(s)}] = \Phi_{\boldsymbol{x}}\big(2^s\mathbb{I}_{\boldsymbol{x}}^{(s)};\Theta_{\boldsymbol{x}}\big). \quad (14)$$

The scale-consistency loss (13)–(14) enforces cross-scale consistency of the estimations at different scales, providing additional regularization for training two INR-based generators.

## 3.3 Progressively coarse-to-fine training for BID

To address the non-linear nature of BID and avoid convergence to trivial solutions, we introduce a progressive learning strategy that trains the INR-based generators at multiple scales. The training process consists of three stages at different scales. Define the fitting terms at different scale $s$ by

$$L_{\mathrm{fit}}^{(s)} = \mathcal{M}_f\big(\boldsymbol{y}\!\downarrow_{2^s}, \Phi_{\boldsymbol{x}}\big(2^s\cdot\mathbb{I}_{\boldsymbol{x}}^{(s)};\Theta_{\boldsymbol{x}}\big)\otimes\Phi_{\boldsymbol{k}}\big(2^s\cdot\mathbb{I}_{\boldsymbol{k}}^{(s)};\Theta_{\boldsymbol{k}}\big)\big). \quad (15)$$

The first stage serves as the initialization, operating at the coarsest scale with the fitting loss $\mathcal{L}_{\mathrm{fit}}^{(S_0)}$. The second stage progressive refines the training from the scale $S_0$ to 0 with both the fitting loss and cross-scale consistency loss. Specifically, at scale $s$, the loss is $\mathcal{L}_{\mathrm{fit}}^{(s)} + \lambda\mathcal{L}_{\mathrm{cross}}^{(s)}$ where $\lambda$ is a weight and $\mathcal{L}_{\mathrm{cross}}^{(s)}$ is the cross-scale consistency loss defined by (13)–(14). The third stage is the final tuning stage at scale 0 with $\mathcal{L}_{\mathrm{fit}}^{(0)}$ only. The training process is summarized in Algorithm 1. IIn our implementation,

---

**Algorithm 1:** Self-supervised progressively coarse-to-fine training for BID

**input**: a blurred image $\boldsymbol{y}$
**output**: an estimated kernel $\boldsymbol{k}^*$ and latent image $\boldsymbol{x}^*$
1: Initializing two generator NNs $\Phi_{\boldsymbol{k}}, \Phi_{\boldsymbol{x}}$ with random weights $\Theta_{\boldsymbol{k}}$ and $\Theta_{\boldsymbol{x}}$;
2: *Initial training*: at the scale $S_0$, training the NNs with only the fitting loss $\mathcal{L}_{\mathrm{fit}}^{(s)}$;
3: *%% Progressively training the NNs*
4: **for** $s \leftarrow S_0$ **to** 0 **do**
$\quad\Big\lfloor$ Training the NNs at the scale $s$ with the loss $\mathcal{L}_{\mathrm{fit}}^{(s)} + \lambda\mathcal{L}_{\mathrm{cross}}^{(s)}$;
5: *Final tuning*: training the NNs at scale 0 with only the fitting loss $\mathcal{L}_{\mathrm{fit}}^{(0)}$;
6. Define $\boldsymbol{k}^* = \Phi_{\boldsymbol{k}}(\mathbb{I}_{\boldsymbol{k}};\Theta_{\boldsymbol{k}}^*)$ and $\boldsymbol{x}^* = \Phi_{\boldsymbol{x}}(\mathbb{I}_{\boldsymbol{x}};\Theta_{\boldsymbol{x}}^*)$.

---

Structural Similarity Index Measure (SSIM) [60] is used for $\mathcal{M}_f(\cdot)$ in (15):

$$\mathcal{M}_f(\boldsymbol{x},\boldsymbol{y}) = (\mu_x^2 + \mu_y^2 + c_1)^{-1}(\sigma_x^2 + \sigma_y^2 + c_2)^{-1}(2\mu_x\mu_y + c_1)(2\sigma_{x,y} + c_2).$$

For $\mathcal{M}_c(\cdot)$ in (13), note that convolution process in (13) can be efficiently computed by by transforming the convolution operation into pointwise multiplication in its discrete Fourier transform (DFT), denoted by $\mathcal{F}(\cdot)$. Thus, we define $\mathcal{M}_c(\cdot)$ in frequency domain with $\ell_1$-norm:

$$\mathcal{M}_c(\boldsymbol{x},\boldsymbol{y}) = \|\mathcal{F}(\boldsymbol{x}) - \mathcal{F}(\boldsymbol{y})\|_1.$$

| | Category | Manmade | Natural | People | Saturated | Text | Average |
|---|---|---|---|---|---|---|---|
| Non-learning | Xu & Jia$^\triangle$ [65] | 19.23/0.654 | 23.03/0.754 | 25.32/0.852 | 14.79/0.563 | 18.56/0.7173 | 20.18/0.708 |
| | Xu *et al.*$^\triangle$ [67] | 17.99/0.597 | 21.38/0.679 | 24.40/0.813 | 14.53/0.538 | 17.64/0.668 | 19.23/0.659 |
| | Zhong *et al.*$^\triangle$ [76] | 17.32/0.556 | 21.07/0.695 | 24.39/0.761 | 14.86/0.602 | 15.86/0.532 | 18.70/0.628 |
| | Michaeli & Irani$^\triangle$ [38] | 17.43/0.418 | 20.70/0.511 | 23.35/0.699 | 14.14/0.491 | 16.23/0.468 | 18.37/0.518 |
| | Pan-DCP $^\triangle$[42] | 18.59/0.594 | 22.60/0.698 | 24.03/0.772 | 16.52/0.632 | 17.42/0.619 | 19.89/0.666 |
| | Yan *et al.*$^\triangle$ [70] | 19.32/0.579 | 23.69/0.678 | 27.01/0.842 | 16.46/0.588 | 18.64/0.689 | 21.02/0.675 |
| | Yang & Ji$^\triangle$ [71] | 19.99/0.599 | 24.33/0.692 | 27.22/0.861 | 17.04/0.605 | 20.35/0.762 | 21.79/0.704 |
| Supervised | DeblurGAN-v2 [25] | 15.93/0.321 | 18.95/0.429 | 21.53/0.694 | 13.79/0.488 | 14.82/0.519 | 17.04/0.490 |
| | Kaufman & Fattal [20] | **18.94/0.517** | **22.05/0.586** | **27.05/0.833** | 15.18/0.599 | **17.85/0.717** | **20.22/0.650** |
| | MIMO-UNet [11] | 15.49/0.301 | 18.36/0.415 | 20.03/0.653 | 13.65/0.473 | 14.26/0.464 | 16.36/0.461 |
| | MPRNet [73] | 15.58/0.309 | 18.56/0.429 | 20.08/0.656 | 13.67/0.478 | 12.83/0.400 | 16.15/0.454 |
| | MPRNet* [73] | 17.39/0.419 | 20.53/0.510 | 22.85/0.673 | 15.35/0.551 | 16.01/0.499 | 18.42/0.531 |
| | Restormer [72] | 15.63/0.324 | 18.55/0.433 | 20.29/0.665 | 13.70/0.499 | 13.40/0.451 | 16.31/0.474 |
| | Restormer*[72] | 17.87/0.453 | 21.07/0.553 | 23.15/0.674 | **15.59/0.550** | 16.67/0.543 | 18.89/0.555 |
| Self-supervised | SelfDeblur$^\triangle$[47] | 20.08/0.538 | 22.50/0.581 | 27.41/0.850 | 16.58/0.654 | 19.06/0.731 | 21.13/0.671 |
| | SelfDeblur[47] | 20.35/0.754 | 22.05/0.709 | 25.94/0.883 | 16.35/0.636 | 20.16/0.779 | 20.97/0.752 |
| | DEBID$^\triangle$[9] | 19.62/0.692 | 24.12/0.807 | 28.23/0.890 | 17.12/0.692 | 19.44/0.711 | 21.71/0.751 |
| | DEBID[9] | 22.14/0.803 | 26.18/0.894 | 31.25/0.923 | 18.43/0.714 | 23.00/0.822 | 24.20/0.831 |
| | MCEM$^\triangle$[33] | 21.01/0.682 | 24.67/0.751 | 28.17/0.863 | 16.63/0.651 | 20.51/0.760 | 22.20/0.741 |
| | MCEM[33] | 23.06/0.751 | 26.00/0.774 | 31.02/0.902 | 17.21/0.679 | 25.46/0.892 | 24.55/0.800 |
| | VDIP$^\triangle$ [18] | 20.97/0.647 | 24.51/0.770 | 27.53/0.862 | 17.18/0.716 | 20.23/0.743 | 22.08/0.747 |
| | VDIP[18] | 22.86/0.868 | 26.18/0.895 | 30.76/0.927 | **18.55/0.727** | 27.24/0.927 | 25.12/0.869 |
| | Ours$^\triangle$ | 21.06/0.698 | 24.70/0.811 | 28.31/0.890 | 16.63/0.655 | 20.67/0.733 | 22.27/0.756 |
| | Ours | **23.24/0.893** | **26.27/0.933** | **31.53/0.944** | 17.76/0.683 | **27.01/0.930** | **25.16/0.879** |

Table 1: Average PSNR/SSIM of the results for Lai *et al.* dataset [26]. The methods marked with $^\triangle$ deblur the image by [23] using the estimated kernel, a standard protocol for evaluating kernel estimation accuracy in BID. The methods marked with * are retrained on the BSD-D dataset [48].

# 4   Experiments

**Implementation details:** The training consists of 5000 iterations across three stages. The first stage operates at the coarsest scale $S_0$ with 500 iterations. The second stage refines training from scale $S_0$ to scale 0, with 500 iterations per scale. The final stage is tuning at scale 0 for the remaining iterations. The NN is trained using the Adam optimizer with a batch size of 1. The initial learning rates for the image and kernel generators are set to $5 \times 10^{-3}$ and $5 \times 10^{-5}$, respectively, decreasing to half their values every 2000 iterations. The weight $\lambda$ in the loss function is set to 0.001 to keep the values of the two loss terms $\mathcal{L}_{\text{cross}}$ and $\mathcal{L}_{\text{fit}}$ in the same order. For comparison, we use results from the literature when available; otherwise, we use pre-trained models or train from the provided code to achieve optimal performance. The code of the proposed method is available on Github.[1]

## 4.1   Evalution of motion deblurring on synthesized datasets

Two metrics, PSNR (peak signal-to-noise ratio) and SSIM, are used for performance evaluation. Following SelfDeblur [47], we compute PSNR/SSIM after finding the best shift between GT and the result to handle shift ambiguity. In the tables, **Bold** in blue indicates the best among all supervised methods, **Bold** in black indicates the best, and underline the second-best among all GT-free methods.

**Synthetic dataset with uniform blurring from Lai *et al.* [26]:** This dataset consists of 100 images categorized into five groups: Manmade, Natural, People, Saturated, and Text, and covers 4 different kernels whose size ranges from $31 \times 31$ to $75 \times 75$. For this dataset, $S_0$ is set to 2 . As the focus of BID is on accurately estimating the blur kernel. Thus, a two-stage evaluation protocol, as outlined by [47], is also used in our test. Note that most supervised methods only generate clear images without kernels, and thus their evaluation is solely on their output images. The competing methods include 7 non-learning methods, 5 supervised methods, and 4 self-supervised methods. For all

---

[1] https://github.com/tjzhang-nus/Deblur-INR

| Metric | Non-learning | | | | | | | Supervised | |
|---|---|---|---|---|---|---|---|---|---|
| | Cho & Lee [12] | Jia *et al.* [67] | Whyte *et al.* [62] | Hirsch *et al.* [16] | Vasu *et al.* [58] | Yan *et al.* [70] | Yang & Ji [71] | SRN [56] | DeblurGAN-v2[25] |
| PSNR | 28.98 | 27.34 | 28.07 | 27.77 | 29.89 | 29.61 | 29.22 | 27.06 | 26.97 |
| MSSIM | 0.933 | 0.796 | 0.848 | 0.852 | 0.927 | N/A | N/A | 0.840 | 0.830 |

| Metric | Supervised | | | | Diffusion | Self-Supervised | | | |
|---|---|---|---|---|---|---|---|---|---|
| | Kaufman& Fattal [20] | MPRNet [73] | MIMO-UNet [11] | Li *et al.* [31] | BlindDPS [13] | SelfDeblur [47] | MCEM [33] | VDIP [18] | Ours |
| PSNR | **30.17** | 26.32 | 25.34 | 26.89 | 24.02 | 25.85 | 30.26 | 29.58 | **30.69** |
| MSSIM | **0.915** | 0.827 | 0.791 | 0.837 | 0.702 | 0.792 | 0.940 | 0.922 | **0.942** |

Table 2: Average PSNR/MSSIM of the results from different methods on the dataset Köhler's dataset [22].

supervised methods, except for [20], we use their models pre-trained on the GoPro dataset [39] with both uniform and non-uniform blurring. For two recent methods, MPRNet [73] and Restormer [72], we also include their models retrained on the BSD-D dataset [48] with only uniform blurring.

The comparison in Tab. 1 shows that our method achieved the best performance. Note that for two supervised methods, MPRNet [73] and Restormer [72], their models trained on a dataset with only uniform blur performs better on testing data with uniform blur, when compared to the ones trained on the dataset with non-uniform blurring. However, this improvement is still not enough to match the performance of our method. The main reason is that existing supervised methods target general blurring and overlook the physics prior of image formation, specifically the convolution model for uniform blurring. As a result, they underperform compared to our self-supervised method, which leverages this prior. This highlights the generalization issues inherent in supervised learning approaches, whose performance heavily depends the correlation degree between the training and testing data. Refer to Appendix F.1 for visual comparisons of different methods.

**Synthetic dataset with modest non-uniform blurring from Köhler *et al.* [22]:** The benchmark dataset Köhler [22] comprises 48 motion-blurred images which blurring is not exactly uniform. This dataset is for evaluating the robustness of BID method to handle modest non-uniform blurring. Totally 17 methods are selected for comparison, and $S_0$ is also set to 2. Following the evaluation protocol in [33], we use the average PSNR and MSSIM (multi-scale SSIM) as the evaluation metrics. Tab. 2 shows that our method outperforms all others in terms of both PSNR and MSSIM, indicating its robustness in handling modest non-uniform blur. Refer to Appendix F.2 for visual comparisons.

### 4.2 Evaluation of motion deblurring on real-world datasets

**Real-world dataset from Lai *et al.*'s [26]:** Lai *et al.*'s real-world dataset [26] consists of 100 real blurred images captured in diverse scenarios using various capturing settings. As no GT images for quantitative evaluation, we present only visual comparisons of many samples. Please refer to Appendix F.3 for visual comparisons of different methods. Overall, our method generates the images with the best visual quality, consistent with its performance on the synthetic datasets.

| Dataset | Metric | Non-learning | | | Supervised | | | Self-supervised | | Ours |
|---|---|---|---|---|---|---|---|---|---|---|
| | | Xu & Jia [65] | Pan-DCP [42] | Hu *et al.* [17] | DeepDeblur [39] | DeblurGAN-v2[25] | Restormer [72] | SelfDeblur [47] | MCEM [33] | |
| RealBlur-J | PSNR | 27.14 | 27.22 | 26.41 | 27.87 | 28.70 | **28.96** | 27.92 | 28.17 | **28.31** |
| | SSIM | 0.8303 | 0.7901 | 0.8028 | 0.8274 | 0.8662 | **0.8790** | 0.8420 | 0.8499 | **0.8511** |
| RealBlur-R | PSNR | 34.46 | 34.01 | 33.67 | 32.51 | 35.26 | **36.19** | 34.49 | 35.46 | **35.62** |
| | SSIM | 0.9368 | 0.9162 | 0.9158 | 0.8406 | 0.9440 | **0.9570** | 0.9270 | 0.9436 | **0.9448** |

Table 3: Average PSNR/SSIM of the results on the RealBlur dataset [48].

**Real-world dataset RealBlur [48] with small/modest blurring:** Our proposed method is also evaluated in another real-world dataset published in [48]: RealBlur-J and RealBlur-R, both containing

980 real-world blurry images. See Tab. 3 for the results. It can be seen that, unlike the experiments on synthetic datasets, there is a gap between the proposed method and supervised methods. This is understandable, as our method cannot access GT images, whereas the supervised methods can. However, our method still outperforms all traditional methods and existing self-supervised methods. Refer to Appendix F.4 for visual comparisons of different methods.

## 4.3 Evaluation on microscopic deconvolution

In microscopic imaging, acquired images often suffer from blur due to optical limitations, out-of-focus elements, specimen motion, and the diffraction limit of light. Following [45], the test dataset consists of 120 images, covering 24 images from the test subsets "Confocal_BPAE_B" and "TwoPhoton_MICE", and includes 3 Gaussian point spread functions (PSFs) and 2 Poisson PSFs as the blur kernels. For the compared supervised methods, Restomer [72] and INIKNet [46], we retrained their models using the microscopic dataset [45]. It can be seen from Tab. 4 that our method outperforms all other self-supervised methods and is comparable to the supervised methods trained on the microscopic dataset. Refer to Appendix F.5 for visual comparisons of different methods.

| PSFs | Supervised | | Diffusion | Self-supervised | | | |
|---|---|---|---|---|---|---|---|
| | Restormer [72] | INIKNet [46] | BlindDPS [13] | SelfDeblur [47] | MCEM [33] | VDIP [18] | Ours |
| Gaussian | 36.32/0.936 | **37.32/0.941** | 26.78/0.661 | 35.52/0.927 | 36.12/0.933 | 35.73/0.932 | **36.65/0.940** |
| Poisson | 40.73/0.958 | **41.56/0.961** | 26.94/0.667 | 39.88/0.950 | 39.26/0.945 | 38.47/0.942 | **40.71/0.960** |

Table 4: Average PSNR/SSIM of the results from different methods on microscopic deconvolution.

| Category | Manmade | Natural | People | Saturated | Text | Average |
|---|---|---|---|---|---|---|
| w/o $\mathcal{L}_{\text{cross}}$ | 21.19/0.778 | 25.84/0.887 | 30.74/0.918 | 17.69/0.682 | 26.75/0.917 | 24.44/0.836 |
| Single-scale | 22.04/0.803 | 25.93/0.890 | 30.33/0.933 | 17.68/0.688 | 24.76/0.886 | 24.14/0.840 |
| w/o Progressive | 20.36/0.742 | 23.91/0.829 | 26.35/0.821 | 17.22/0.675 | 22.88/0.857 | 22.14/0.790 |
| INR/CNN as $\Phi_k/\Phi_x$ | 16.62/0.370 | 26.21/0.865 | 28.83/0877 | 17.03/0.668 | 23.34/0.776 | 23.01/0.767 |
| MLP/INR as $\Phi_k/\Phi_x$ | 19.88/0.661 | 19.47/0.479 | 26.77/0.718 | 16.07/0.667 | 15.66/0.537 | 19.17/0.521 |
| MLP/CNN as $\Phi_k/\Phi_x$ | 15.87/0.331 | 19.20/0.430 | 23.32/0.578 | 15.58/0.627 | 16.57/0.481 | 18.19/0.470 |
| Multi-scale by direct $\downarrow$ | 18.64/0.616 | 22.61/0.722 | 25.95/0.781 | 16.64/0.601 | 20.65/0.861 | 20.89/0.716 |
| Ours | **23.24/0.893** | **26.27/0.933** | **31.53/0.944** | **17.76/0.683** | **27.01/0.930** | **25.16/0.879** |

Table 5: Ablation study of the proposed method in terms of of PSNR/SSIM.

## 4.4 Ablation study

The ablation study is conducted on the Lai *et al.* dataset [26]. The results are shown in Tab. 5

**Effectiveness of self-supervised cross-scale consistency loss:** To evaluate the gain from the proposed cross-scale consistency loss functions, we retrain the NN using only the fitting loss $\mathcal{L}_{\text{fit}}$ for each scale, *i.e.*, without (w/o) $\mathcal{L}_{\text{cross}}$. Tab. 5 shows an average gain of about 0.72 dB in PSNR with the cross-scale consistency loss, demonstrating its effectiveness.

**Effective of progressive coarse-to-fine training:** We first examine the effectiveness of multi-scale training by performing the training only at the original scale. Tab. 5 shows about a 1 dB loss in PSNR, indicating multi-scale training's contribution. To evaluate the progressive training strategy, we train the NN with the sum of all loss functions at three scales, which resulting in a 3dB loss. This clearly indicates that progressive training is critical for effectively utilizing the multi-scale scheme. The reason is that the loss function at the coarser scale emphasizes lower frequencies since the image at coarser scale retains low but loses high frequencies. Thus, a NN trained on the sum of loss functions across 3 scales focuses more on low frequencies than one trained only at the finest scale, which fits both low and high frequencies. Therefore, a scale-progressive training scheme is more effective than a joint multi-scale loss. In scale-progressive training, the estimation at the coarser scale provides

a strong initialization for the finer scales. The final result is obtained by applying the fitting term exclusively at the finest scale.

**INR (coordinate NN) vs. MLP/CNN (image-to-image NN)** This ablation study is to evaluate the necessity of using INR for re-parametrizatrion of image and/or kernel, instead of using the NN that maps an image to an image. In the study, we separately replace our INR-based kernel/image generator with MLP/CNN based representation adopted in [47], whose down-sampled versions of images/kernels at different scales are generated by standard down-sampling process.

**Direct down-sampling vs. using multi-scale grid in INR**: To verify the benefit of resolution-free property of INR, we also consider generating multi-scale versions of image/kernel by standard down-sampling using bilinear interpolation, on their high resolution version from INPs, same to a CNN-based representation. This is referred to as "Multi-scale by direct $\downarrow$". Tab. 5 shows that the performance of multi-scale by standard down-sampling process is very poor. Clearly multi-scale representation by standard downsampling process is not as effective as resolution-free INR.

### 4.5 Comparison of computational effciency

The computational effciency of the propoposed method is compared to three most related self-supervised BID methods. The results are reported in terms of running time, number of parameters, and memory usage, when processing a $256 \times 256$ image with a $31 \times 31$ blur kernel on an NVIDIA 3090 RTX GPU. See Table 6 for the details. It can be seen that the proposed method achieves a balance between computational cost and deblurring performance.

| Methods | SelfDeblur [47] | MCEM [33] | VDIP [18] | Ours |
|---|---|---|---|---|
| Running time (s) | 219.71 | 226.31 | 245.04 | 213.02 |
| Number of parameters (k) | 3427.2 | 2409.0 | 3523.2 | 2342.4 |
| Memory usage (GB) | 6.31 | 1.27 | 9.19 | 1.82 |

Table 6: Model complexity comparison.

### 4.6 More details, studies, experiments and visual comparisons in the appendix

The appendix includes (1) ablation study on setting scale $S_0$, (2) details on hyper-parameter settings, (3) details on NN architecture, (4) more comparison on running time; (5) visualization of some sample results, (6) visualization of intermediate results, and (7) experiments on Levin *et al.*'s dataset [28].

## 5 Discussion and Conclusion

This paper introduces a self-supervised method for BID, which does not require GT images. Leveraging on the resolution-free representation of INR for image/kernel, we propose a self-supervised cross-scale consistency loss function and a progressive coarse-to-fine training strategy. The proposed method significantly outperforms existing methods in extensive experiments.

There are two limitations of the proposed self-supervised method. The first is the computational cost for processing a large number of images, as the method requires training the model for each individual sample. A potential solution is to explore the usage of the proposed techniques in meta-learning or testing-time adaptation. This would allow the proposed technique to rapidly adapt a pre-trained model instead of to train the NN from scratch. The second limitation is that the proposed method is only applicable to handle uniform blurring, as it relies on the convolution model. Extending this approach to handle non-uniform blur will be another important direction for future research.

## Acknowledgments

Yuhui Quan would like to acknowledge the support from National Natural Science Foundation of China under Grant 62372186, Natural Science Foundation of Guangdong Province under Grants 2022A1515011755 and 2023A1515012841, and Fundamental Research Funds for the Central Universities under Grant x2jsD2230220. Hui Ji would like to acknowledge the support from Singapore MOE Academic Research Fund (AcRF) Tier 1 Research Grant A-8000981-00-00.

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

# 6 Appendix

Below we provide additional details and results which are not presented in the main manuscript.

## A Proof of Proposition 1

*Proof.* The proof is based on the convolution theorem and the properties of the DFT. Without loss of generality, we assume that the image $\boldsymbol{x}$ and the kernel $\boldsymbol{k}$ are of size $2M \times 2N$. Let $\boldsymbol{x}{\downarrow}_2$ and $\boldsymbol{k}{\downarrow}_2$ denote the down-sampled images of $\boldsymbol{x}$ and $\boldsymbol{k}$, respectively. Let $\mathcal{F}$ denote the discrete Fourier transform (DFT). Recall that for a signal $\boldsymbol{z} \in \mathbb{R}^{2M \times 2N}$, its DFT is defined as:

$$\mathcal{F}(\boldsymbol{z})[m, n] = \sum_{\ell=0}^{2M-1} \sum_{j=0}^{2N-1} \boldsymbol{z}[\ell, j] e^{-i2\pi \frac{m}{2M} \ell} e^{-i2\pi \frac{n}{2N} j}.$$

Then, for a down-sampled version of the signal $\boldsymbol{z}{\downarrow}_2$, whose DFT is

$$\mathcal{F}(\boldsymbol{z}{\downarrow}_2)[m, n] = \sum_{\ell=0}^{M-1} \sum_{j=0}^{N-1} \boldsymbol{z}[2\ell, 2j] e^{-i2\pi \frac{m}{M} \ell} e^{-i2\pi \frac{n}{N} j}.$$

The DFTs of the down-sampled signals $\boldsymbol{x}{\downarrow}_2$ and $\boldsymbol{k}{\downarrow}_2$ can be expressed as follows: For each frequency $[m, n]$, we have

$$\begin{aligned}
\mathcal{F}\left(\boldsymbol{x}{\downarrow}_2\right)[m, n] = \frac{1}{4}(&\mathcal{F}(\boldsymbol{x})[2m, 2n] + \mathcal{F}(\boldsymbol{x})[2m + M, 2n] \\
&+ \mathcal{F}(\boldsymbol{x})[2m, 2n + N] + \mathcal{F}(\boldsymbol{x})[2m + M, 2n + N]),
\end{aligned} \tag{16}$$

$$\begin{aligned}
\mathcal{F}\left(\boldsymbol{k}{\downarrow}_2\right)[m, n] = \frac{1}{4}(&\mathcal{F}(\boldsymbol{k})[2m, 2n] + \mathcal{F}(\boldsymbol{k})[2m + M, n] \\
&+ \mathcal{F}(\boldsymbol{k})[2m, 2n + N] + \mathcal{F}(\boldsymbol{k})[2m + M, 2n + N]),
\end{aligned} \tag{17}$$

and

$$\begin{aligned}
\mathcal{F}\left(\boldsymbol{y}{\downarrow}_2\right)[m, n] = \frac{1}{4}(&\mathcal{F}(\boldsymbol{y})[2m, 2n] + \mathcal{F}(\boldsymbol{y})[2m + M, n] \\
&+ \mathcal{F}(\boldsymbol{y})[2m, 2n + N] + \mathcal{F}(\boldsymbol{y})[2m + M, 2n + N]),
\end{aligned} \tag{18}$$

By convolution theorem, for $\boldsymbol{y} = \boldsymbol{x} \otimes \boldsymbol{k}$, we have that

$$\mathcal{F}(\boldsymbol{y})[m, n] = \mathcal{F}(\boldsymbol{x})[m, n] \cdot \mathcal{F}(\boldsymbol{k})[m, n]$$

Thus,

$$\mathcal{F}\left(\boldsymbol{y}{\downarrow}_2\right)[m, n] = \frac{1}{4}\left(\sum_{\ell=0}^{1} \sum_{j=0}^{1} \mathcal{F}(\boldsymbol{x})[2m + \ell M, 2n + jN] \cdot \mathcal{F}(\boldsymbol{k})[2m + \ell M, 2n + jN]\right). \tag{19}$$

Applying convolution theorem on $\boldsymbol{x}{\downarrow}_2 \otimes \boldsymbol{k}{\downarrow}_2$ again, we have

$$\mathcal{F}(\boldsymbol{x}{\downarrow}_2 \otimes \boldsymbol{k}{\downarrow}_2)[m, n] = \mathcal{F}(\boldsymbol{x}{\downarrow}_2)[m, n] \cdot \mathcal{F}(\boldsymbol{k}{\downarrow}_2)[m, n]. \tag{20}$$

Plugging in (16) and (17) into the above equation, we have

$$\mathcal{F}(\boldsymbol{x}{\downarrow}_2 \otimes \boldsymbol{k}{\downarrow}_2)[m, n] = \frac{1}{16}\left(\sum_{\ell=0}^{1} \sum_{j=0}^{1} \mathcal{F}(\boldsymbol{x})[2m + \ell M, 2n + jN] \cdot \sum_{i=0}^{1} \sum_{j=0}^{1} \mathcal{F}(\boldsymbol{k})[2m + \ell M, 2n + jN]\right). \tag{21}$$

Define 3 filters $\boldsymbol{g}_1, \boldsymbol{g}_2, \boldsymbol{g}_3$ by

$$\begin{aligned}
\boldsymbol{g}_1[m, n] &= (-1)^m \boldsymbol{k}[m, n] \\
\boldsymbol{g}_2[m, n] &= (-1)^n \boldsymbol{k}[m, n] \\
\boldsymbol{g}_3[m, n] &= (-1)^{m+n} \boldsymbol{k}[m, n]
\end{aligned} \tag{22}$$

Then, theirs DFTs are given by

$$
\begin{aligned}
\mathcal{F}(\boldsymbol{g}_1)[m,n] &= \mathcal{F}(\boldsymbol{k})[m+M,n] \\
\mathcal{F}(\boldsymbol{g}_2)[m,n] &= \mathcal{F}(\boldsymbol{k})[m,n+N] \\
\mathcal{F}(\boldsymbol{g}_3)[m,n] &= \mathcal{F}(\boldsymbol{k})[m+M,n+N]
\end{aligned}
\tag{23}
$$

Then, the expression of the right-hand side of Equation (21) can be rewritten as:

$$
\mathcal{F}(\boldsymbol{x}\!\downarrow_2 \otimes \boldsymbol{k}\!\downarrow_2)[m,n] = \frac{1}{4}\left( \mathcal{F}\left((\boldsymbol{x}\otimes\boldsymbol{k})\!\downarrow_2\right)[m,n] + \sum_{d=1}^{3}\mathcal{F}\left((\boldsymbol{x}\otimes\boldsymbol{g}_d)\!\downarrow_2\right)[m,n] \right)
\tag{24}
$$

By convolution theorem, we have then

$$
(\boldsymbol{x}\!\downarrow_2 \otimes \boldsymbol{k}\!\downarrow_2) = \frac{1}{4}\left( \left((\boldsymbol{x}\otimes\boldsymbol{k})\!\downarrow_2\right) + \sum_{d=1}^{3}\left((\boldsymbol{x}\otimes\boldsymbol{g}_d)\!\downarrow_2\right) \right),
$$

The proof is complete. $\qquad\square$

# B  Ablation study on performance impact by different maximum scale $S_0$

In this ablation study, we investigate how different choices of scale $S_0$ impact the performance on motion deblurring tasks. The experiments are conducted using Lai *et al.*'s dataset. We compare results using the same framework with different $S_0$ values: single-scale, two-scale, three-scale, and four-scale (i.e., $S_0 \in \{0,1,2,3\}$). Each setting undergoes a total of 5000 training epochs, with each stage except the last one consisting of 500 training epochs.

As shown in Tab. 7, multi-scale frameworks consistently outperform single-scale setups. Among the multi-scale configurations, the three-scale model demonstrates the most significant improvement. However, the four-scale model shows only a slight improvement over the single-scale setup. This minimal gain can be attributed to the initial stages of the four-scale framework, where the target image, resized to 1/8 of the original size, is far from the true image. This highly erroneous initialization causes more harm than benefit, hindering the NN's convergence to the true image.

| Category | Manmade | Natural | People | Saturated | Text | Average |
|---|---|---|---|---|---|---|
| Single-scale ($S_0 = 0$) | 22.04/0.803 | 25.93/0.890 | 30.33/0.933 | 17.68/0.688 | 24.76/0.886 | 24.14/0.840 |
| Two-scale ($S_0 = 1$) | 22.87/0.866 | 25.89/0.888 | 30.54/0.921 | **17.89/0.693** | 26.72/0.919 | 24.78/0.857 |
| Three-scale ($S_0 = 2$) | **23.24/0.893** | **26.27/0.933** | **31.53/0.944** | 17.76/0.683 | **27.01/0.930** | **25.16/0.879** |
| Four-scale ($S_0 = 3$) | 22.89/0.869 | 25.04/0.866 | 30.21/0.899 | 17.71/0.699 | 25.21/0.903 | 24.21/0.847 |

Table 7: Scale variation study on the proposed architecture in terms of PSNR/SSIM on the dataset Lai *et al.* [26]. **Bold** for best performers and underline for second-best performers.

# C  Implementation Details

## C.1  Hyper-parameter settings for different Datasets

The following Tab. 8 summarizes the hyper-parameters, specifically the range of kernel sizes and the coarsest scale $S_0$ used in our model across various datasets. These parameters are selected based on the specific characteristics of each dataset, including image size.

## C.2  Details of image generators

Please see Tab. 9 for the details of the architectures used for image generative network $\Phi_{\boldsymbol{x}}$. In Tab. 9, the "CBR" denotes the subblock for three successive layers: Conv2d+BatchNorm2d+ReLU. The "CBR/Down" means modifying the Conv2d layer in "CBR" with stride $(2,2)$, which performs downsampling operations. The "Up" means the upsampling layer with bilinear interpolation. The "Cat(m,n)" denotes the concatenation of the outputs from layers No.$m$ and No.$n$.

| Dataset | Kernel Size Range | Coarsest Scale $S_0$ |
|---|---|---|
| Lai *et al.*'s dataset [26] | $31 \times 31 - 75 \times 75$ | 2 |
| Köhler *et al.*'s dataset [22] | $27 \times 27 - 131 \times 131$ | 2 |
| Lai *et al.*'s real dataset [26] | $27 \times 27 - 99 \times 99$ | 2 |
| Realblur datasets [48] | $27 \times 27 - 79 \times 79$ | 2 |
| Microscopic datasets [45] | $5 \times 5 - 13 \times 13$ | 1 |
| Levin *et al.*'s dataset [28] | $11 \times 11 - 27 \times 27$ | 1 |

Table 8: Kernel size ranges and coarsest scale settings for various datasets.

| | No. | Block | Channels | No. | Block | Channels | No. | Block | Channels | No. | Block | Channels |
|---|---|---|---|---|---|---|---|---|---|---|---|---|
| | | Encoder | | | | | | Decoder | | | | |
| | 1 | Encoding by Eq. 8 | 64 | 9 | CBR/Down | 128 | 17 | Up | – | 25 | CBR | 128 |
| | 2 | CBR | 128 | 10 | CBR | 128 | 18 | Cat(14,17) | 144 | 26 | Up | – |
| | 3 | CBR/Down | 128 | 11 | CBR | 16 | 19 | CBR | 128 | 27 | Cat(5,26) | 144 |
| $\Phi_x$ | 4 | CBR | 128 | 12 | CBR/Down | 128 | 20 | Up | – | 28 | CBR | 128 |
| | 5 | CBR | 16 | 13 | CBR | 128 | 21 | Cat(11,20) | 144 | 29 | Up | – |
| | 6 | CBR/Down | 128 | 14 | CBR | 16 | 22 | CBR | 128 | 30 | Cat(2,29) | 128 |
| | 7 | CBR | 128 | 15 | CBR/Down | 128 | 23 | Up | – | 31 | CBR$\times$2 | 128 |
| | 8 | CBR | 16 | 16 | CBR | 128 | 24 | Cat(8,23) | 144 | 32 | Sigmoid | C |

Table 9: Architecture details of image generative NN $\Phi_x$ in the proposed method

# D    Additional comparison of computational cost

Tab. 10 reports the running times for processing a $256 \times 256$ image with a $31 \times 31$ blur kernel, comparing severe existing BID methods, including those beyond self-supervised approaches. The inference time is measured using an NVIDIA 3090 RTX GPU. Unlike supervised methods, which require considerable time for model training but is very fast on processing images using pre-trained model, our approach involves training a model for each specific image that needs processing. This makes our method similar to traditional iterative methods, focusing on a case-by-case basis rather than relying on a pre-trained model. The running time for processing an image is comparable to many iterative optimization methods and existing self-supervised learning solutions. One of future works would be on how to reducing this per-image training time, possibly through the implementation of meta learning or testing-time model adaption.

| | Non-learning | | | | Supervised | | Diffusion-based | Self-Supervised | | | |
|---|---|---|---|---|---|---|---|---|---|---|---|
| | Sun *et al.* [54] | Jin *et al.* [19] | Yan *et al.* [70] | Yang & Ji [71] | MPRNet [73] | Restormer [72] | BlindDPS [13] | SelfDeblur [47] | MCEM [33] | VDIP [18] | Ours |
| Time(s) | 113.98 | 35.84 | 1242.97 | 354.03 | 0.21 | 0.25 | 286.55 | 219.71 | 226.33 | 245.04 | 213.02 |

Table 10: Time comparison with existing BID methods when processing a $256 \times 256$ image

# E    Broader impacts

 The proposed self-supervised learning method for deblurring images has the potential to impact a wide range of applications, including surveillance security, scientific research, and digital media restoration. By improving the accuracy and clarity of images, our research can facilitate deeper insights and more effective interventions in these fields.

In surveillance, higher clarity images can enhance public safety by providing more detailed visual information. However, this also raises concerns about privacy and the potential for mass surveillance. In scientific research, improved image quality can lead to better data and more significant discoveries, though there is a risk that overly processed images could misrepresent the original data. In digital media restoration, while the technique helps preserve cultural heritage, it also poses the risk of altering historical records.

Despite these possible concerns, our goal is to contribute to enhancing image clarity in critical areas such as public safety, scientific research, and cultural preservation. We emphasize the responsible application and continuous improvement of this technology to mitigate potential risks and maximize its positive impact.

# F  Visual comparisons of the results in the experiments

## F.1  Visual comparisons on Lai *et al.*'s dataset

In this section, we visualize the results of different methods on some examples from Lai *et al.*'s dataset [26], which is known for its severe blurring effects. Figs. 1 - 6 demonstrate that our method consistently produces results with sharper details and fewer artifacts compared to existing methods, showcasing its effectiveness in addressing significant blur challenges. In contrast, supervised learning methods trained on external datasets yield poor quality results, highlighting the limited generalization performance of supervised approaches when dealing with complex real-world blurring.

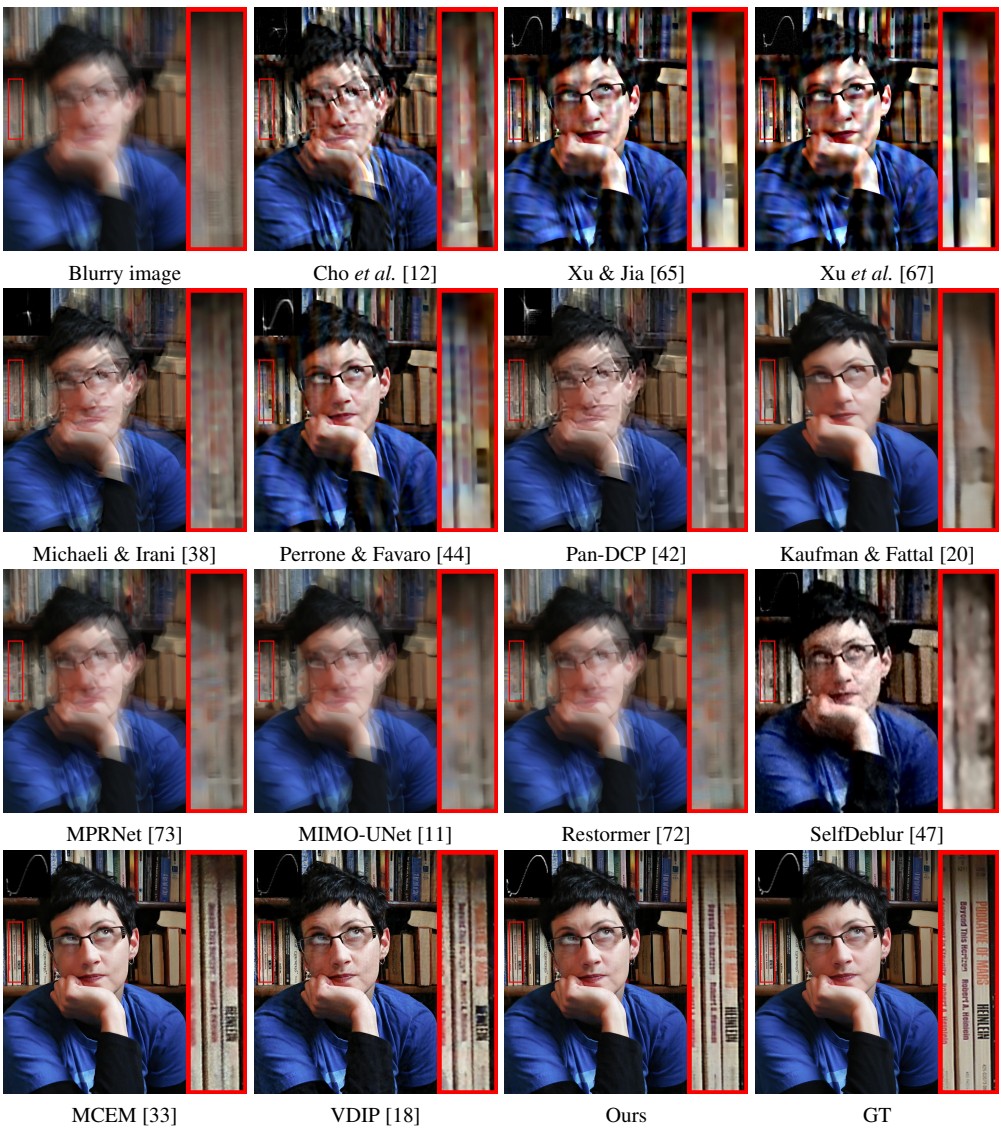

Figure 1: Visual results on the dataset of Lai *et al.* [26]

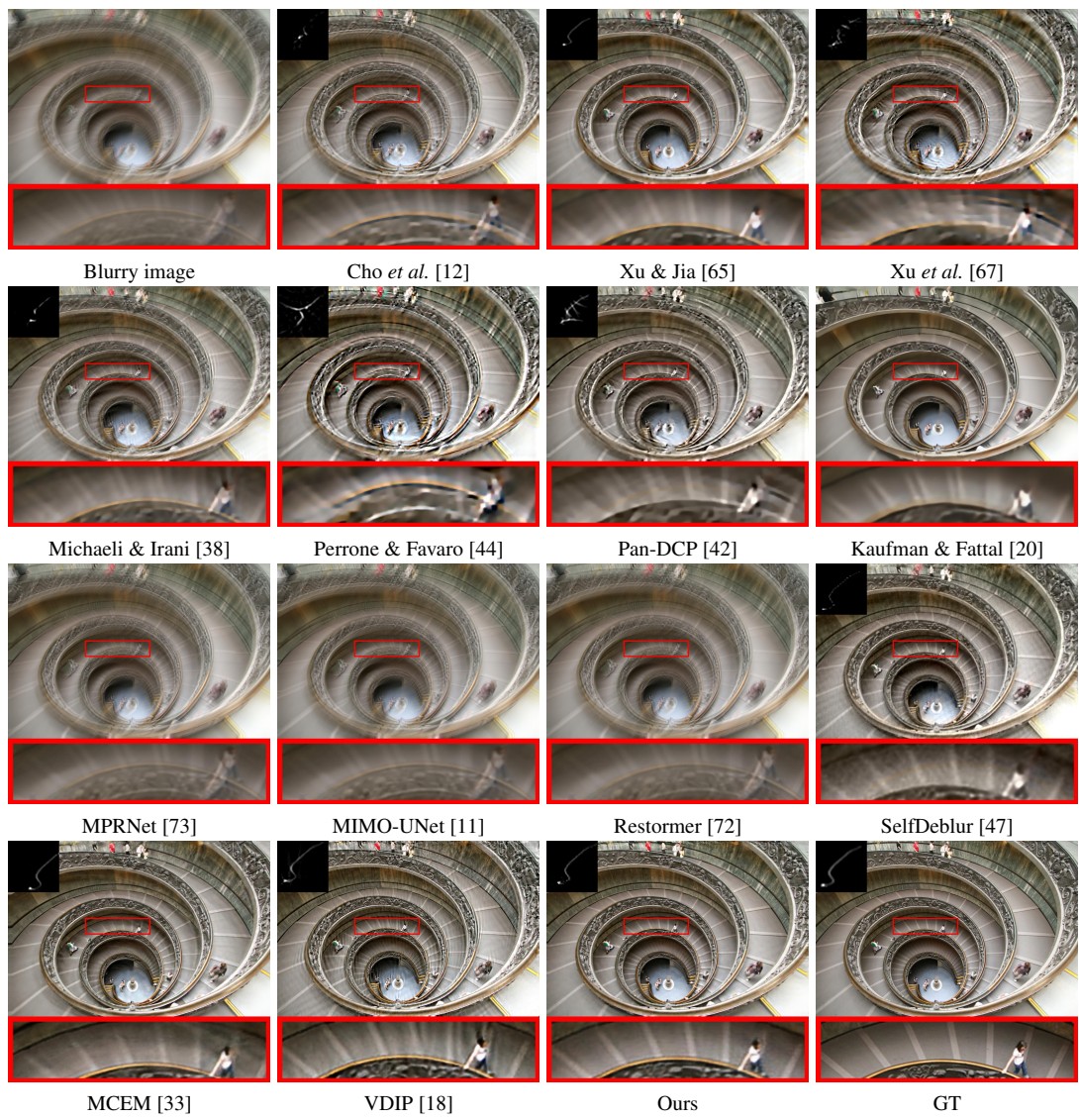

Figure 2: Visual results on the dataset of Lai *et al.* [26]

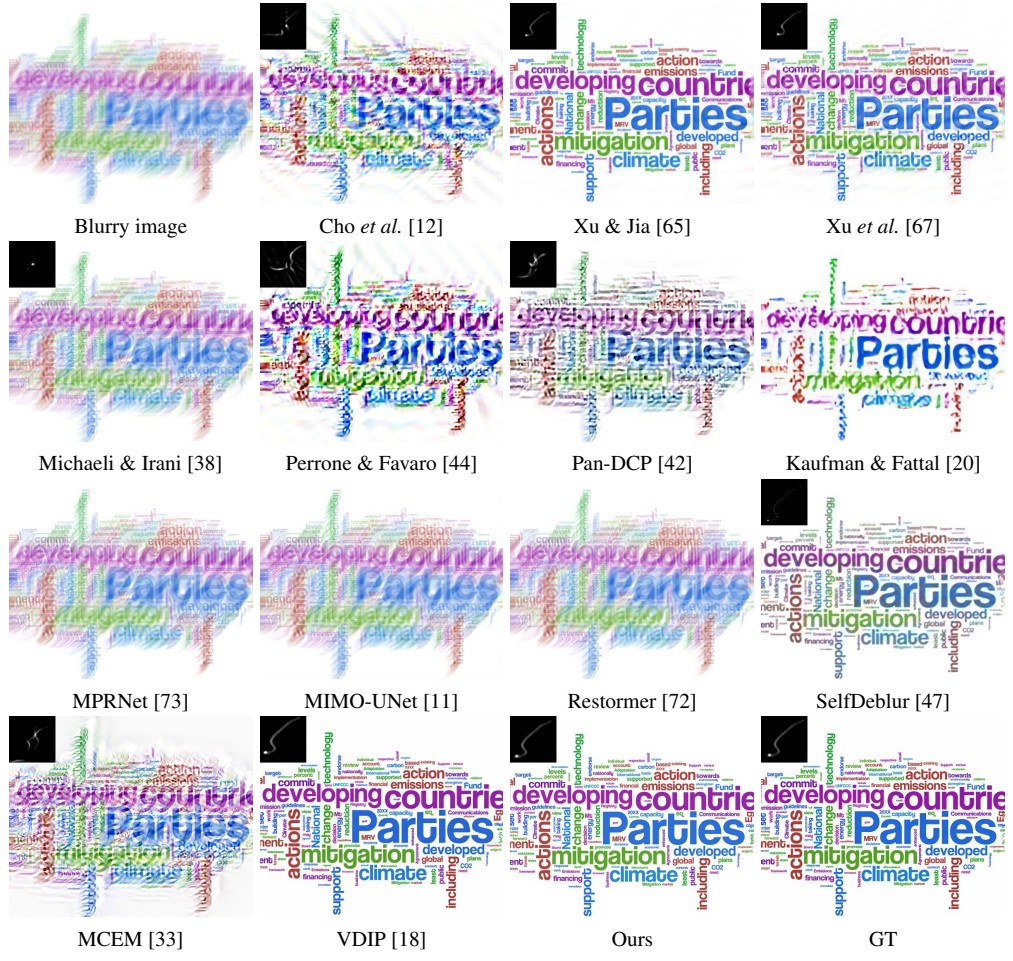

Figure 3: Visual results on the dataset of Lai *et al.* [26]

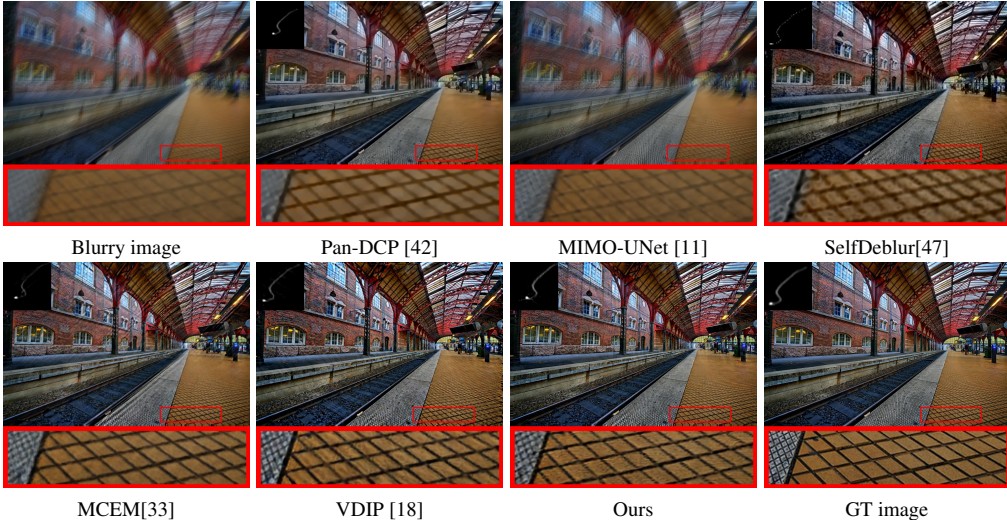

Figure 4: Visual comparison on the dataset of Lai *et al.* [26]

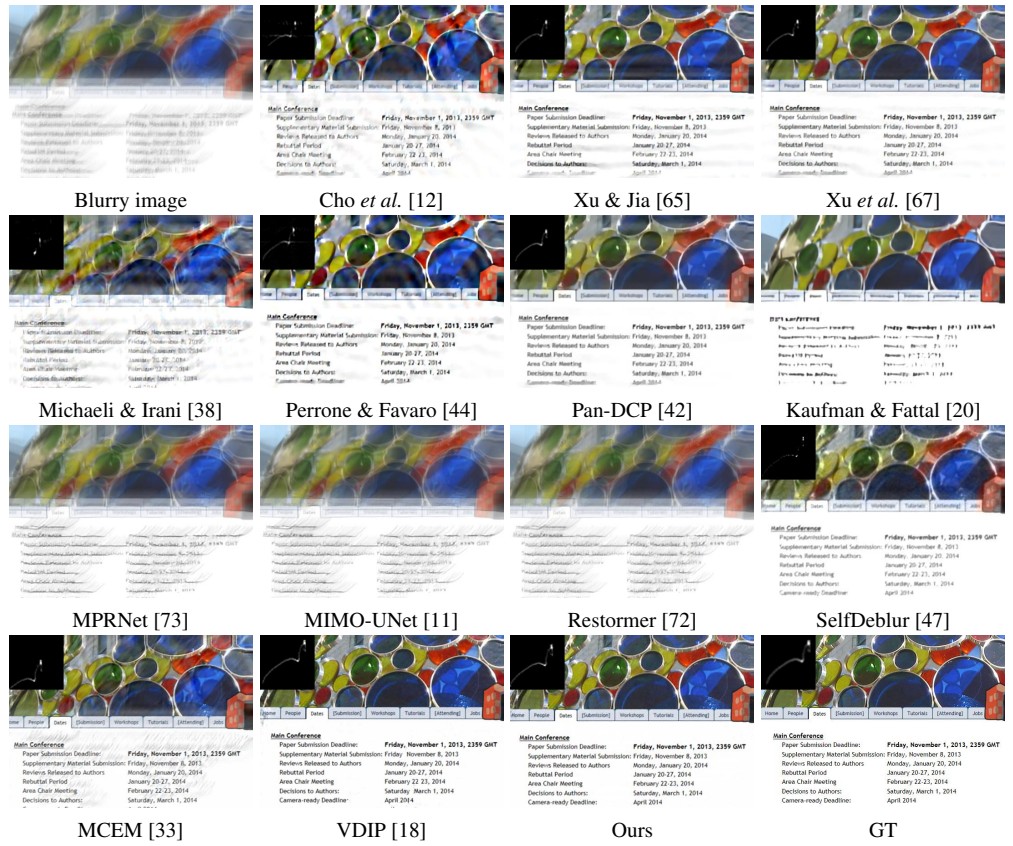

Figure 5: Visual results on the dataset of Lai *et al.* [26]

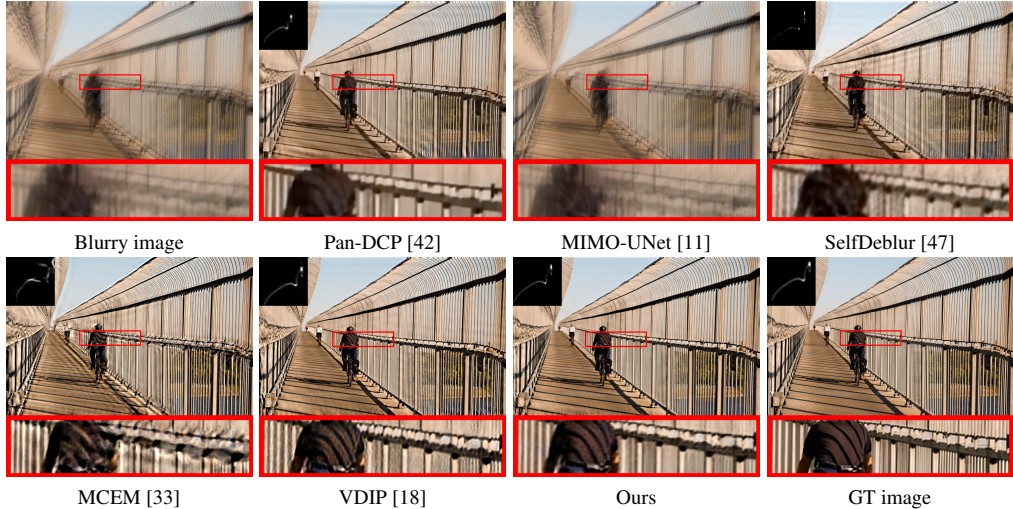

Figure 6: Visual comparison on the dataset of Lai *et al*. [26]

## F.2  Visual comparisons on Köhler *et al.*'s dataset

In this section, we display the visual results on Köhler *et al.*'s dataset [22], which is characterized by its non-uniform blurring effects. From Fig. 7-Fig. 8, our method demonstrats satisfying performance on this challenging dataset, effectively handling the intricate and variable blurring effects. Furthermore, we compare our results with those obtained using the Diffusion-based method, BlindDPS [13], revealing its difficulty in adapting to blurring effects and image types that deviate from its pre-trained model's training data.

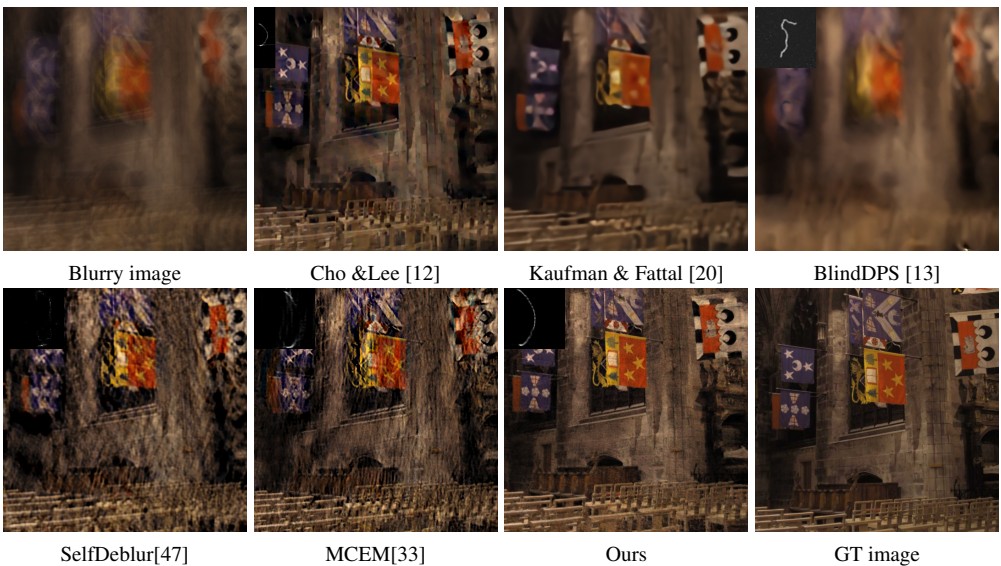

Figure 7: Visual comparison on the dataset of Köhler *et al.* [22]

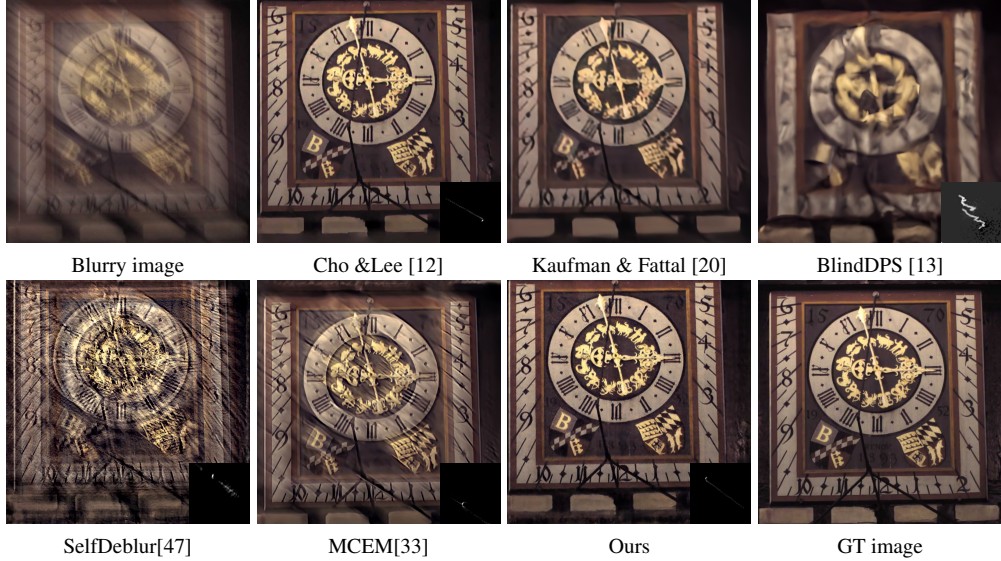

Figure 8: Visual comparison on the dataset of Köhler *et al.* [22]

## F.3  Visual comparisons on Lai *et al.*'s real dataset

Fig. 9 illustrates a comparison between our method and other competitive self-supervised methods on Lai *et al.*'s real-world data, where our method consistently generates higher quality images.

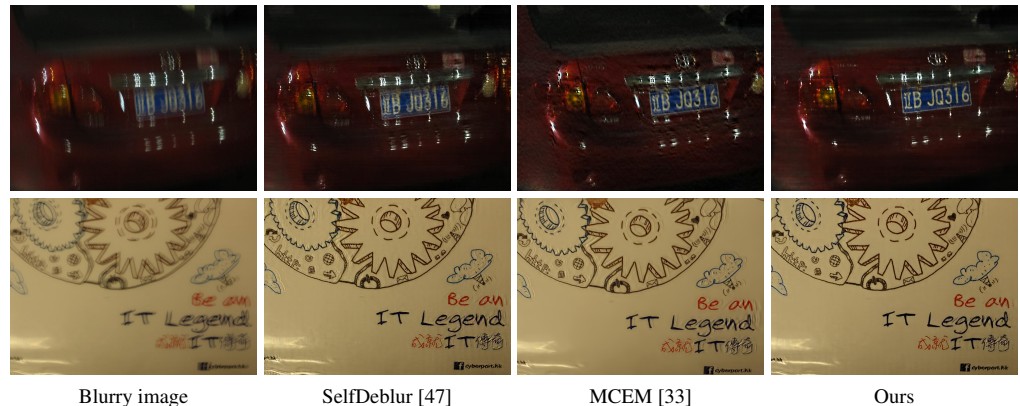

| Blurry image | SelfDeblur [47] | MCEM [33] | Ours |

Figure 9: Visual comparison on the real dataset of Lai *et al.* [26]

## F.4 Visual comparisons on RealBlur dataset

In this section, we present a visual comparison of the results from state-of-the-art supervised learning methods [73, 11, 72] and our approach on the RealBlur dataset. Although our method achieves slightly lower average PSNR and SSIM values across the entire dataset compared to these representative supervised learning methods, the visual evidence presented in Fig. 10 demonstrates that our results are often sharper. In other words, the gap between our method and the supervised methods is smaller in terms of visual quality than indicated by the quantitative metrics.

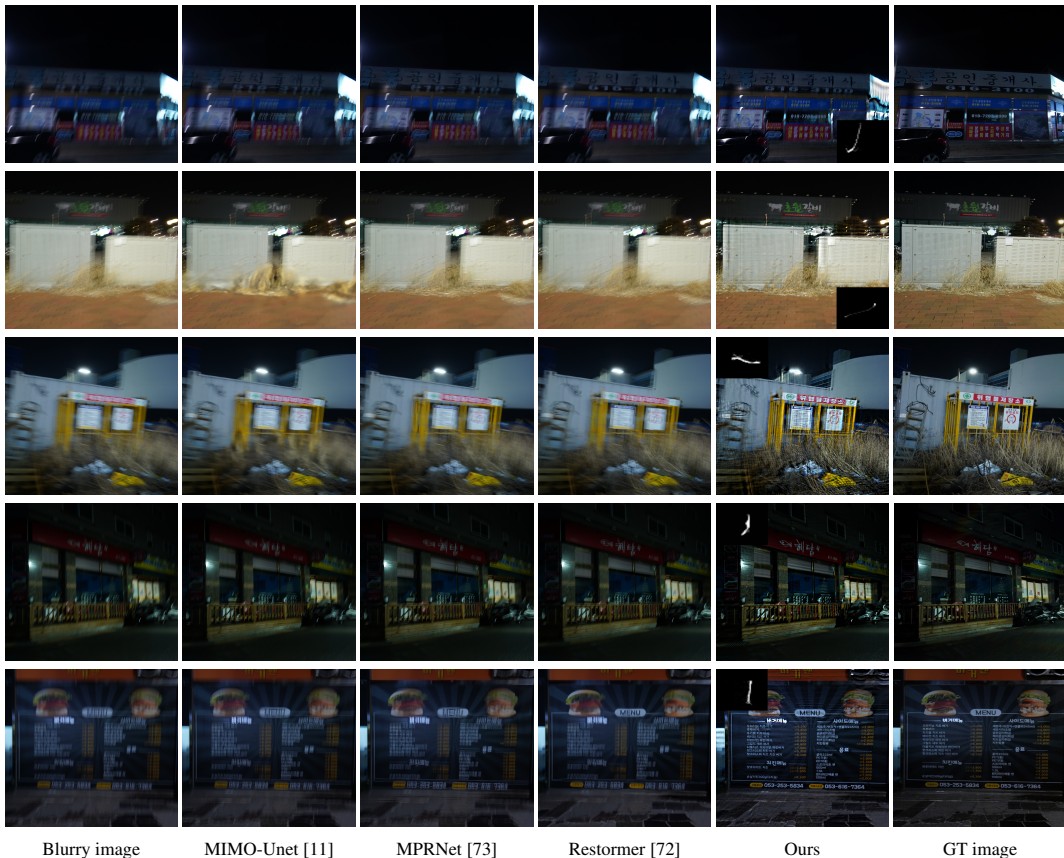

| Blurry image | MIMO-Unet [11] | MPRNet [73] | Restormer [72] | Ours | GT image |

Figure 10: Visual comparison with supervised methods on challenging cases from RealBlur-J [48].

### F.5 Visual comparisons on Microscopic dataset

This section provides a comparison between our method and other methods in the task of microscopic deconvolution. Our approach consistently restores images with more texture and details.

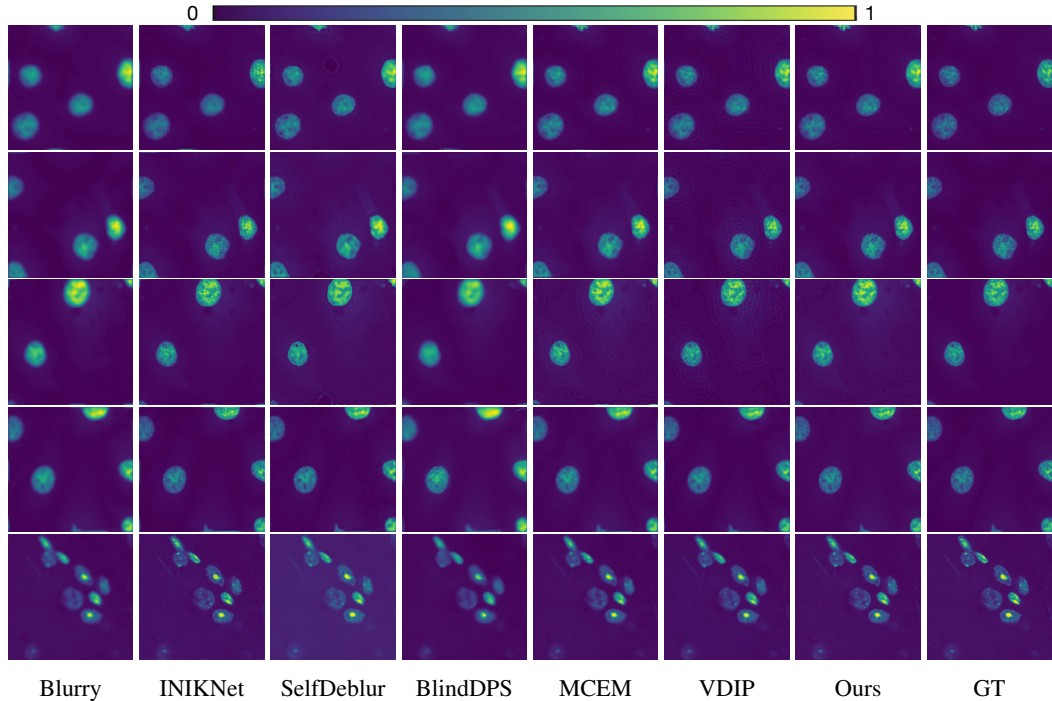

| Blurry | INIKNet | SelfDeblur | BlindDPS | MCEM | VDIP | Ours | GT |

Figure 11: Visual comparison on the microscopic deconvolution. All images are originally grayscale but a different colormap is used to better highlight the differences among the various reconstructions

## G  Visual comparison on the ablation study

This section provides a comprehensive visual comparison for the ablation study in Fig. 12, illustrating the impact of various configurations on our model's performance. These visuals serve as a supplement to the detailed quantitative results discussed in the main text, found in Sec. 4.4.

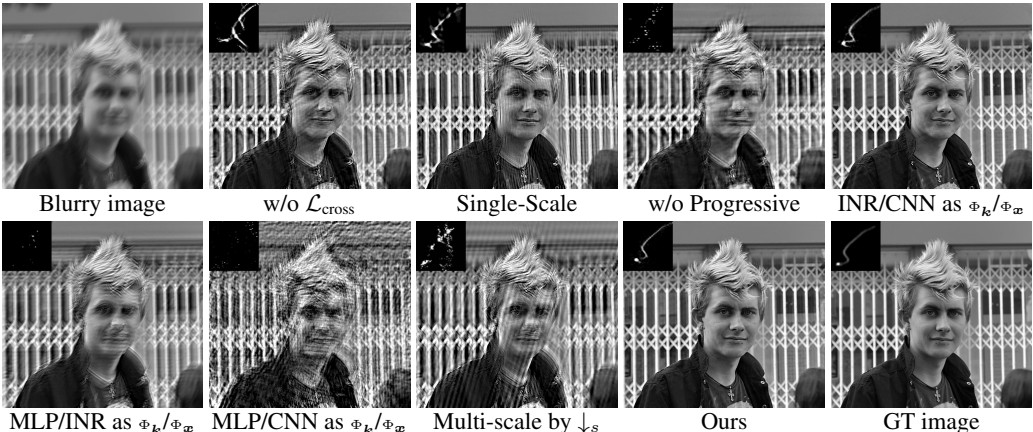

Figure 12: Visual comparison on the ablation study.

# H   Visualization of Intermediate Results

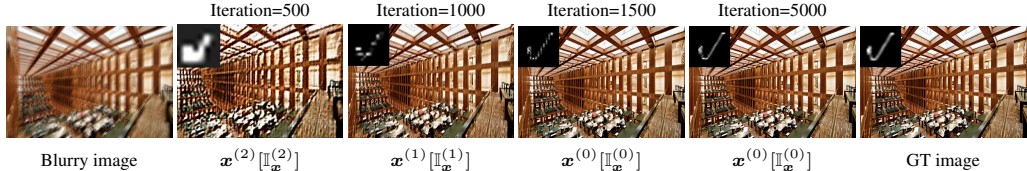

Figure 13: Intermediate results of estimated blur kernel and latent image at end of each stage.

Apart from validating the efficacy of our approach through the quality of the final results, it is also interestting to see the intermediate status of the proposed method. To this end, we use "manmade_01" from Lai *et al.*'s dataset [26] as a case study, visualizing intermediate results to better understand the progress through different stages to the final output. As shown in Figure 13, in the first stage, we obtain an initial representation at the smallest scale during initialization. In the second and third stages, we achieve results at larger scales with finer details. In the final stage, the neural network builds on the groundwork laid by the preceding stages to refine the output, ultimately generating the desired clear image.

# I   Experiments on Levin *et al.*'s dataset [28] with small kernel size.

Levin *et al.*'s dataset contains 32 images generated by convolving 4 clear images using 8 motion-blur kernels and adding Gaussian white noise with s.t.d. $1\%$. The size of these kernels is small, ranging from $11 \times 11$ to $27 \times 27$. Following [47], besides PSNR and SSIM, Error Ratio [30] is also used as a quantitative metric. From Tab. 11, our approach performs well across all three metrics. For visual comparisons, please see Figs. 14- 15 where our results more closely resemble the GT image, with fewer artifacts and more details.

| Metric | Non-learning | | | | | | | |
|---|---|---|---|---|---|---|---|---|
| | known **k** | Krishnan *et al.* [24] | Cho&Lee [12] | Levin *et al.* [29] | Xu & Jia [65] | Sun *et al.* [54] | Zuo *et al.* [78] | Pan-DCP [42] |
| PSNR | 34.53 | 29.88 | 30.57 | 30.80 | 31.67 | 32.99 | 32.66 | 32.69 |
| SSIM | 0.949 | 0.866 | 0.896 | 0.909 | 0.916 | 0.933 | 0.933 | 0.928 |
| Error Ratio | 1.000 | 2.452 | 1.711 | 1.772 | 1.489 | 1.284 | 1.250 | 1.255 |

| Metric | Supervised | | | Diffusion-based | Self-Supervised | | | |
|---|---|---|---|---|---|---|---|---|
| | SRN* [56] | MPRNet [73] | Restormer [72] | BlindDPS [13] | SelfDeblur [47] | MCEM [18] | VDIP [33] | Ours |
| PSNR | 23.43 | 26.21 | **27.78** | 14.91 | 33.07 | 32.81 | 33.12 | **33.74** |
| SSIM | 0.712 | 0.795 | **0.838** | 0.367 | 0.931 | 0.927 | 0.929 | **0.938** |
| Error Ratio | 6.086 | 3.050 | **2.633** | 21.96 | 1.196 | 1.273 | 1.188 | **1.185** |

Table 11: Average PSNR/SSIM of the results from different methods on the dataset Levin *et al.*'s dataset [28]. The method marked with * is retrained with synthesized datasets using eight blur kernels in the dataset of Levin *et al.* [28].

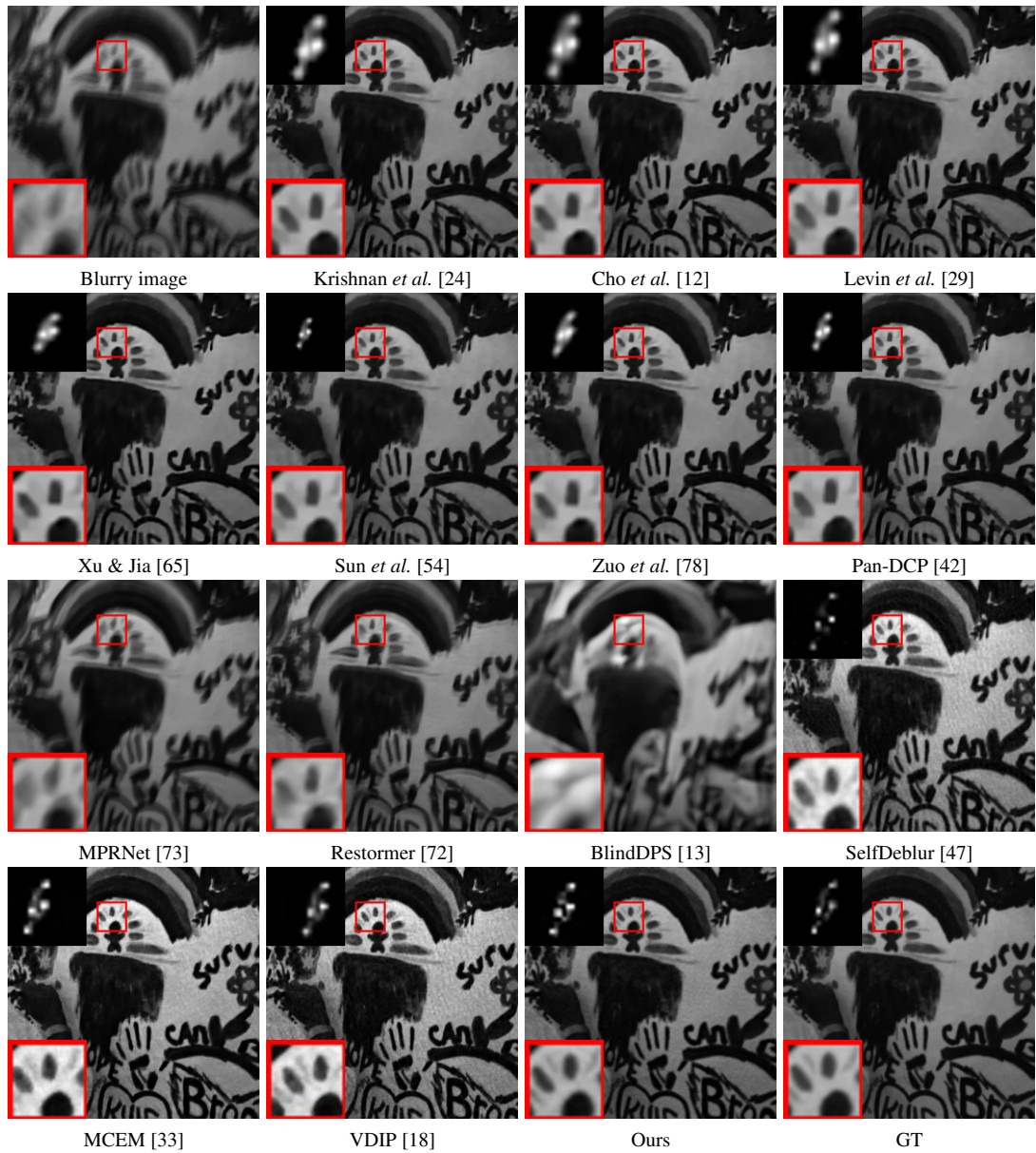

Figure 14: Visual comparison on the dataset of Levin *et al.* [28]

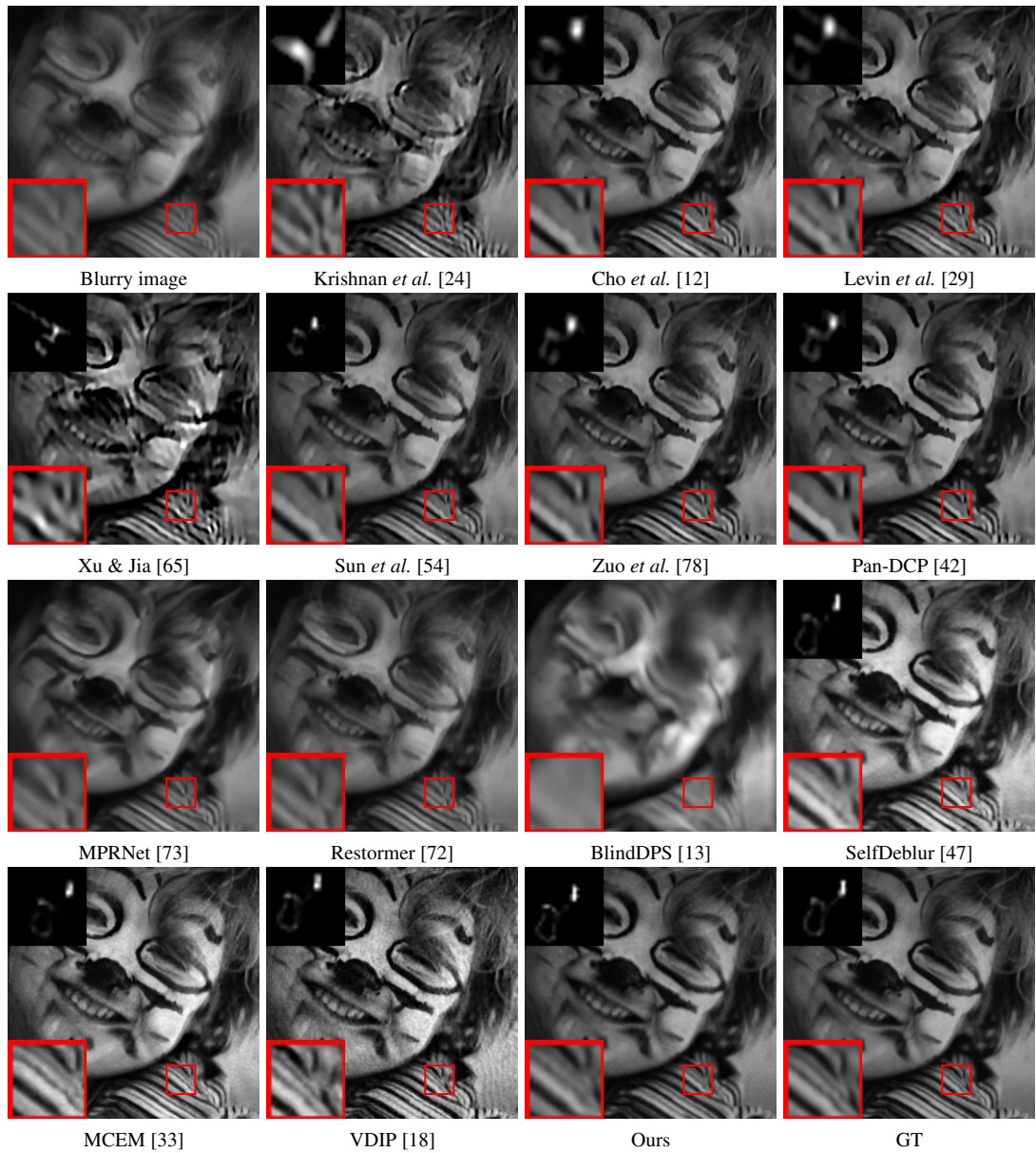

Figure 15: Visual comparison on the dataset of Levin *et al.* [28]

