# OpenReview forum: "Cross-Scale Self-Supervised Blind Image Deblurring via Implicit Neural Representation"
_NeurIPS.cc/2024/Conference — NeurIPS 2024 poster_

### Official Review · Reviewer_UEmh · 2024-06-14

**Soundness:** 2
**Presentation:** 3
**Contribution:** 2
**Rating:** 5
**Confidence:** 4

**Summary:**

This paper proposes a blind image blurring method which reparametrizes the latent images the blurring kernel by the implicit neural representations (INRs). In addition, the authors also propose a cross-scale consistency loss. The authors validate the effectiveness of their method on several datasets.

**Strengths:**

1. The cross-scale consistency loss sounds an interesting idea.
2. The presentation of this paper is good.

**Weaknesses:**

1) The idea of using a deep neural network for latent images and blurring kernel reparametrizatrion is not new. The current proposed method is very similar to the one in ref1. But I do not see the authors have discussed the differences between the proposed method and the method in Ref1. Thus, I doubt the novelty of this proposed method.

2) The current design, the authors use different network architectures to generate images and kernel. But how to control or balance the complexity between latent images generator and the kernel generator?  The authors need to make it more clear why use these specific network design for images and kernel.

3) If the batch size is 1, will it be better to use other norms instead of batch norm?

4) The authors should also compare their method with the one in Ref1 as the Ref1 seems to be the more recent one compared to other self-supervised learning method.

5) The authors should also show how the generated kernel looks like. Is the generated kernel feasible? Is it close to the ground-truth kernel?

Re1: Zhuang, Zhong, et al. "Blind image deblurring with unknown kernel size and substantial noise." International Journal of Computer Vision 132.2 (2024): 319-348.


=========
score has been revised after reading rebuttals.

**Questions:**

I only see very limited novelty of this paper. The proposed idea is similar to the one in Ref1. The only contribution I can see is the cross-scale consistency loss. I list the details in the [Weaknesses].

Re1: Zhuang, Zhong, et al. "Blind image deblurring with unknown kernel size and substantial noise." International Journal of Computer Vision 132.2 (2024): 319-348.

**Limitations:**

Yes.

---

> ### Author Rebuttal · Authors · 2024-08-06
>
> Thanks the reviewer for the comments. See below for our responses to the concerns and questions.
>
> **[W1]** *The idea of using a deep neural network for latent images and blurring kernel reparametrizatrion is not new. The current proposed method is very similar to the one in ref1. But I do not see the authors have discussed the differences between the proposed method and the method in Ref1. Thus, I doubt the novelty of this proposed method.*
>
> Thanks for the reference. We never claimed that using NN-based representation for image/kernel is our  contribution. NN-based image/kernel representation started with DIP, and there has been an extended list of such self-supervised methods for image restoration, including [1]. Our main contribution is to further improve the performance of such an approach for BID by introducing a self-supervised cross-scale consistency loss, along with a progressive training scheme, both built on the resolution-free property of INRs.
>
> Our work is very different from [1]. While both utilize INRs, the main differences include:
> (a) We propose a cross-scale consistency constraint for regularization of BID, while [1] uses L1/L2 norm regularization, early stopping, and other tricks for regularization.
> (b) [1] does not consider a multi-scale-based training scheme, whereas we propose a multi-scale progressive training scheme.
> (c) The INP is used the same way as CNN in [1], while we utilize the resolution-free property of INPs for implementing the proposed constraint and progressive training.
>
> [1] Zhong et al. "Blind image deblurring with unknown kernel size and substantial noise". IJCV, 2024.
>
> ---
>
> **[W2]** *The current design, the authors use different network architectures to generate images and kernel. But how to control or balance the complexity between latent images generator and the kernel generator? The authors need to make it more clear why use these specific network design for images and kernel:*
>
> Similar to most existing works on self-supervised BID,  our approach employ different architecture and model sizes for the kernel and image. Given that image size is generally much larger than kernel size, the model size for representing an image needs to be considerably larger than that for a kernel. Therefore, within our coordinate NN, we use a U-Net with parameter no. 2292(k) for the image, and a three-layer MLP with parameters no. 50(k) for the kernel. This design choice is consistent with previous methods [2, 3].
>
> [2] Ren et al.  "Neural blind deconvolution using deep priors."  CVPR, 2020.
>
> [3] Chen et al. "Self-supervised blind image deconvolution via deep generative ensemble learning."  TCSVT, 2023
>
> ---
>
> **[W3]** *If the batch size is 1, will it be better to use other norms instead of batch norm?*
>
> Thank for pointing it out. We agree that the usefulness of batch BN becomes limited when the batch size is 1, which also happens in existing works [2,4,5], We initial experiments on adopting LayerNorm shows no impact in the results.  We will elaborate it more in revision.
>
> [4] Li et al. "Self-supervised blind motion deblurring with deep expectation maximization." CVPR, 2023.
>
> [5] Ulyanov et al. "Deep image prior." CVPR. 2018.
>
> ---
>
> **[W4]** *The authors should also compare their method with the one in Ref1 as the Ref1 seems to be the more recent one compared to other self-supervised learning method.*
>
> There is no code available in the public domain for [1], and we also didn't receive the code for evaluation after  contacting the authors of [1]. Additionally, the experiments conducted in [1] do not include any dataset with the same configuration as evaluated in this paper. Considering the complex regularization used in [1], it is challenging to make a fair comparison.
>
> ---
>
> **[W5]** *The authors should also show how the generated kernel looks like. Is the generated kernel feasible? Is it close to the ground-truth kernel?*
>
> We would like to point out that, we have included visual results of the generated kernels in the Appendix, displayed in the top-left corner of the corresponding images. The estimated kernels are very close to the ground-truth kernels. Please refer to our Appendix, specifically Fig. 1--Fig. 12, for these visual comparisons.

---

> > ### Author Response · Authors · 2024-08-13
> >
> > Dear reviewer,
> >
> > Thank you again for your time and effort in reviewing our work. We have carefully addressed all the concerns and questions you raised, including the point regarding the specific paper you mentioned for discussion.
> >
> > We understand that the review process is demanding. However, as the deadline for discussion on the response is approaching, we wanted to ensure that our responses are clear and adequately address your concerns and questions. Any further clarification or specific questions related to our response would be invaluable to us. Thank you once again for your time and for considering our responses.

---

> > ### Comment · Reviewer_UEmh · 2024-08-13
> >
> > Thank you so much for your rebuttal. I have carefully read the rebuttal as well as the comments and discussions. I raised my rating accordingly based on all the updated information (rebuttal, other available reviews, and discussions).

---

### Official Review · Reviewer_PNcB · 2024-07-02

**Soundness:** 2
**Presentation:** 3
**Contribution:** 2
**Rating:** 5
**Confidence:** 4

**Summary:**

This paper introduces a self-supervised method for BID that does not require GT images. By leveraging an exact relationship among the blurred image, latent image, and blur kernel across consecutive scales, this paper propose an **effective cross-scale consistency loss** implemented by representing the image and kernel with **implicit neural representations** (INRs), whose resolution-free property **enables consistent yet efficient computation for network training at multiple scales**. The experimental part  verifies that proposed method outperforms some existing self supervised methods on several datasets.

**Strengths:**

- The paper proposes a new method on the existing self-supervised niform BID task, which is complete and sound. The main contribution is the cross-scale estimation consistency constraint, and provides a complete theoretical proof of the rationality of the regularization term. The cross-scale consistency is used to train progressive scale IRNs from convergencing to trivial solutions.  which has a certain originality and provides an inspiring new analytical way for BID.
- The paper conducts in-depth theoretical and experimental analysis on uniform-based deconvolution, verifies the superiority of the proposed method through experimental comparison on two types of motion blurred image synthetic data and real data, and combines sufficient ablation experiments to verify the impact of each module of the framework on performance.
- Aside from a few details, the submission is clearly written and generally well organized, and the supplementary materials are accompanied by a large amount of quantitative and visualization materials.
- The authors are careful and honest about evaluating both the strengths and weaknesses。

**Weaknesses:**

- The INRs framework and progressive learning mechanism on which the paper is based are common in image restoration, the degree of innovation is average except for cross-scale estimation consistency constraint.
- With the development of BID or the larger blind image restoration area, the degradation settings of different methods are somewhat different. When describing the paper’s setting (Eq.1), in addition to comparison methods, introducing the existing blurring degradation settings (uniform kernel/non-uniform kernel/implicit modeling) and their differences and application value is necessary. The discussion of this part is missing, which causes readers/researchers confused by different method settings and applicability.
- The related work section of the paper does not fully cite and introduce recent learning-based self-supervised or frequency estimation degradation and some important works, such as KernelGAN(NeurIPS2019), FCA(AAAI2021), S2K(NeurIPS2021), etc. These solutions also involve deconvolution and propose estimation methods.
- Some descriptions of the method principles or experimental settings are vague or problematic, see Questions.

**Questions:**

- There are some confusing points in Tab.1: (1) The results of the supervised method are of little reference value for they are trained on non-convolutional modeling datasets(GoPro). (2) According to the ablation experiment, even without the main contribution (1) mentioned in line 73, the PSNR(24.44) is almost the same as the second best method(MCEM) (24.55). The uniform blur setting is inconsistent with some existing methods (such as the non-uniform modeling used by MCEM). Are those self-supervised method retrained for comparison?
- The proposed uniform blurring method is better than MCEM in the non-uniform deblurring experiment (Tab.2), which is a bit contradictory. Is this an advantage brought by individual sample training rather than the method itself?
- In the multi-scale training process, what is the effect of x and k estimated in the previous scale on the next scale? From the image, it can be understood that coarse-to-fine is getting clearer. How does k change in this process? In addition, in Tab.5, the average PSNR of training only at the original scale is two points higher than that of training with at three scales, indicating that the optimization direction of BID at different scales is inconsistent. In this case, why does training with progressive scales bring better performance?
- According to the existing conclusions in the BID field in lines 94-98 of the paper, the second contribution point of the paper summarized in lines 107-108 is not important. In addition, the use of multi-scale coarse-to-fine can alleviate the ill-posed problem of multiple solutions. There is no analysis or literature references provided though some work has been done before indeed.
- In Lines 191-192, it claims the role of the cross-scale estimation consistency is to limit the ambiguous solution of BID. Though a rational analysis of the approach and ablation experiments are provided, readers may wonder what is the motivation for designing this regularization term and its actual impact in BID process. If there is some intuitive analysis or explanation, it will be easier for them to accept.
- For the standard downsampling implemented by the coordinate index in line 165, is interpolation  need  when the image size is not even? Is the downsampling used in the ablation experiment to compare image-to-image NNs done in the same way or with more common interpolation methods such as bicubic or bilinear?

**Limitations:**

The authors have  adequately addressed the limitations and potential negative societal impact of their work as follows:
- Computational cost for processing a large number of images, as the method requires training the model for each individual sample.
- The proposed method is only applicable to handle uniform blurring, as it relies on the convolution model.

Considering the practical application value and the single-image training and reasoning method proposed, the paper should compare the computational cost of the single-image reasoning results and the comparison with existing methods.

As mentioned above, it is necessary to add discussions about different setting and it is confusing that proposed method performs better than MCEM in non-uniform setting.

---

> ### Author Rebuttal · Authors · 2024-08-06
>
> Thanks for the comments. Please see below for the responses.
>
> **[W1]** *The INRs framework and progressive learning mechanism on which the paper is based are common. The degree of innovation is average except for cross-scale estimation consistency constraint:*
>
> We agree that the cross-scale consistency constraint is our main contribution. Our innovation with INRs lies in being the first that exploits their resolution-free property for image restoration. It is critical for utilizing cross-scale consistency, as shown in our ablation study. Direct down-sampling is ineffective. Our second contribution is progressive training, which, though established in other areas, is first effectively implemented in self-supervised deep learning for BID using INRs.
>
> ---
> **[W2]**  *The discussion of degradation settings of different methods is missing*
>
> Thanks for the suggestion. There are typically 3 settings: uniform blurring, non-uniform blurring of static scenes, and non-uniform blurring of dynamic scenes with moving objects. Due to space limitations, we focus on uniform blurring. In the revision, we will add more details on the methods regarding their degradation settings and practical applications.
>
> ---
> **[W3]** *The missing reference, such as KernelGAN(NeurIPS2019), FCA(AAAI2021), S2K(NeurIPS2021), etc. They also propose estimation methods:*
>
>
> Thanks for the references. These aim for super-resolution (SR), not blind deblurring. While both estimate the blur kernel, they have different inputs: low-resolution images (SR) versus high-resolution images (BID). They also deal with different blurring kernels: approximately isotropic (SR) versus strongly anisotropic (BID). While they might benefit each other in basic ideas, the techniques in one cannot be easily applied to the other. We will add these references in the revision.
>
> ---
> **[Q1]** *The results of the supervised method are of little reference value for they are trained on non-convolutional modeling datasets:*
>
> Please refer to the global response for the response.
>
> ---
> **[Q2]**  *Even without the contribution (1) mentioned in line 73, the PSNR is almost the same as the second best method (MCEM):*
>
> Note that we have 2 contributions: (1) cross-scale consistency constraint and (2) progressive multi-scale training. Without progressive training, the performance is 2dB lower than MCEM. With only progressive training, ours does not surpass MCEM either. The cross-scale consistency constraint provides a further 0.7 dB gain, letting ours to noticeably outperform MCEM. This justifies the importance of combining both (1) and (2).
>
> ---
> **[Q3]** *Are those self-supervised method retrained for comparison?*
>
>  Most referenced self-supervised methods do not use pre-training; they train a NN directly on the test sample to get results. MCEM uses a piece-wise convolution model with codes for both non-uniform and uniform blurring. The results are either cited from their papers (SelfDeblur[50] and DEBID[11]) or generated using their codes (MCEM[32]).
>
> ---
> **[Q4]** *Why the proposed one outperforms MCEM, and is this an advantage  by individual sample training rather than the method itself?*
>
> Both MCEM and ours train individual NNs on each sample. The Köhler dataset's non-uniformity is modest and can be approximated with a convolution model. For general non-uniform datasets like Lai et al.'s, our method does not outperform MCEM. We will clarify this in the revision.
>
> ---
> **[Q5]** *Regarding the effect of x and k:*
>
> The estimation, $x$ and $k$, from the coarser scale, provides the initialization of the finer scale. Like  the image,  the kernel gets closer to the truth as the process progresses. Please refer to Appendix J for details.
>
> ---
> **[Q6]** *Regarding why training with progressive scales bring better performance:*
>
> The loss function at the coarser scale emphasizes lower frequencies since the image at coarser scale retains low but loses high frequencies. Thus, a NN trained on the sum of loss functions across 3 scales focuses more on low frequencies than one trained only at the finest scale, which fits both low and high frequencies. This motivates our scale-progressive training scheme over the joint multi-scale loss. In progressive training, the estimate from the coarser scale serves as a good initialization for the estimation at the final scale. The final result is achieved by using only the fitting term at the finest scale.
>
> ---
> **[Q7]** *Regarding the importance of the second contribution, and the references:*
>
> While the coarse-to-fine approach has been effective in traditional iterative methods (e.g.[1,2]), we are the first to apply it in deep self-supervised learning for BID. Direct down-sampling in deep learning is ineffective, as shown in Table 5. Our contribution is using resolution-free INR to implement this coarse-to-fine scheme successfully.
>
> [1] Li et al. "Unnatural l0 sparse representation for natural image deblurring". CVPR, 2013
>
> [2] Yang and Ji. "A variational EM framework with adaptive edge selection for blind motion deblurring". CVPR, 2019.
>
> ---
> **[Q8]** *Regarding the role of the cross-scale estimation consistency:*
>
> First, the cross-scale consistency provides additional constraints for regularization. Secondly, it ensures that the estimate from the coarse scale is more accurate than using the existing one,  ${x} \otimes {k}) \downarrow_2 \neq {x} \downarrow_2 \otimes {k} \downarrow_2$, which provides a better initial estimate for the final scale.
>
> ---
> **[Q9]** *Regarding the interpolation and down-sampling*
>
> In our INR-based approach, interpolation isn't needed for uneven image sizes, as INR uses a coordinate NN providing a resolution-free kernel/image. For comparison experiments on image-to-image NNs, we used bilinear downsampling to compute consistency loss and generate low-scale images. We will clarify this in the revision.
>
> ---
> **[Limitation]** *Computational cost of reasoning*
>
> Please refer to the global response for the reply.

---

> > ### Comment · Reviewer_PNcB · 2024-08-13
> >
> > Thanks for replying and clearing up most of my concerns. From the additional experiments provided, self-supervised methods are indeed superior to supervised methods in cross-domain deblurring. It is an important step to make fair experiments clear about the degradation settings of different methods and compare them accordingly. I hope the paper will provide additional explanations in this regard. Based on the rebuttal, I give the final vote.

---

### Official Review · Reviewer_ihYN · 2024-07-12

**Soundness:** 3
**Presentation:** 2
**Contribution:** 2
**Rating:** 6
**Confidence:** 4

**Summary:**

This paper proposed a self-supervised method for blind image deconvolution (BID). The main idea is to introduce the implicit neural representation (INR) technique for representing both the blur kernel and the image, such that they can be parameterized at different scales using a single model. With such a INR representation, supervision can be enforced in a cross-scale way to enhance the efficacy of the model. Extensive experiments have been conducted to demonstrate the effectiveness of the proposed method.

**Strengths:**

1. Introducing INR to the studied BID problem is new and brings benefits in cross-scale representation.
2. Various datasets have been considered in experiments, which makes the results convincing.

**Weaknesses:**

1. The presentation of this manuscript should be improved. For example, a large part of Section 1.2 can be put into Section 3.
2. Experiments are insufficient in the following sense:
- A recent self-supervised BID method [1] is not compared.
- The cross-scale supervision training strategy can also be applied other self-supervised methods by with simple downscale operation, which is not tested.
- The use of the loss functions is not well justified. For example, it is not clear what the performance could be if other more popular ones are used, such as L2 and L1 loss.
- Model complexity comparison in addition to the running time in Appendix D should be provided.

References:

[1] D. Huo et al. Blind Image Deconvolution Using Variational Deep Image Prior. IEEE Transactions on Pattern Analysis and
Machine Intelligence, 45(10): 11472-11483, 2023.

**Questions:**

1. To my knowledge, the performance of fully supervised methods highly depends on the training data. So I would like to know how the supervised methods were trained on the Lai, Kohler, and Levin datasets, since only test images are contained in them.
2. The method is constructed based on the uniform blurring assumption, and thus it should be discussed why it can perform well on the Kohler dataset which focuses on non-uniform blurring.
3. Typos: In Line 31, $\delta,y=k\otimes x$ should be $\delta\otimes y=k\otimes x$.

**Limitations:**

The authors discussed the limitations of the method, which are reasonable.

---

> ### Author Rebuttal · Authors · 2024-08-06
>
> Thank you for your valuable comments. Please see below for our responses.
>
> **[W1]** *The presentation of this manuscript should be improved.*
>
> Thanks for the feedback, we will improve the organization by moving some parts of Sec. 1.2 into Sec. 3.
>
> ---
>
> **[W2.1]** *Comparison with Recent Self-Supervised BID Method VDIP [Dong et al., PAMI, 2023]:*
>
> Thanks for the reference. Please see Table 1 for the comparison in PSNR(dB)/SSIM over 4 datasets. The results show that ours generally outperforms VDIP on all datasets, especially on real-world datasets. We also present visaul comparison in our attanched PDF file. It can easily observe that the proposed method can better restore the blurry images.  Besides, we compare the model complexity between two in Table 2, which shows that our method has significant smaller memory usage and model size.
>
> *Table 1: Average PSNR(dB)/SSIM of the results from VDIP and Ours*
> | Method        | Lai et al.’s Dataset | Köhler’s Dataset | Levin et al.’s Dataset | Microscopic Dataset|
> |:-------------:|:------------------:|:----------------:|:----------------------:|:-------------------:|
> | VDIP   | 25.12 / 0.869      | 29.58 / 0.922    | 33.12 / 0.929          |    37.10 / 0.937 |
> | Ours    | **25.16 / 0.879**      | **30.69 / 0.942**    | **33.74 / 0.938**          | **38.25 / 0.948** |
>
> *Table 2: Model complexity of VDIP and ours*
> | Method| Params (k) |Memory Usage (GB)|Running Time (s)|
> |:-------------:|:------------------:|:----------------:|:----------------------:|
> | VDIP   |3523.2|9.19|245.04|
> | Ours   |2342.4|1.82|213.02|
>
> We will include this comparison in the revision.
>
> ---
>
> **[W2.2]** *The cross-scale supervision training strategy can also be applied to other self-supervised methods by with simple downscale operation:*
>
> Thanks for comments. The study you suggested is already done in the ablation study from lines 276-280, "INR (coordinate NN) vs. MLP/CNN (image-to-image NN)," with results reported in the "MLP/CNN" row of Table 5 of the paper. In this study, we replaced INR with a CNN (image-to-image) and used bi-linear down-sampling for the cross-scale loss, which actually is what you suggested for the existing work "SelfDeblur[50]". The results indicate its much worse performance than the  original single-scale version of "SelfDeblur[50]" with PSNR: 18.19 (cross-scale) vs. 20.97 (single-scale).  It shows that, the resolution-free property of INRs is crucial for effectively exploiting the potential of cross-scale consistency loss. We will clarify this in the revision.
>
> ---
>
> **[W2.3]** *The use of the loss functions is not well justified. For example, it is not clear what the performance could be if other more popular ones are used, such as L2 and L1 loss:*
>
> The metric used in our work is not the focus. Thus, our method employs SSIM Loss, aligning with the most relevant previous works, such as MCEM[32], SelfDeblur[50], and VDIP. The latter two initially trained the NN with L2 loss and switched to SSIM loss after 500/2000 iterations. See Table 3 below for experiments replacing SSIM Loss with L2 Loss, which shows around a 1dB decrease in PSNR across different methods.
>
> *Table 3: Average PSNR(dB)/SSIM of the results from different methods on the Lai et al.'s dataset.*
>    | Loss Function | SelfDeblur  | VDIP  | MCEM  | Ours |
>    |:-------------:|:----------:|:--------:|:----:|:----:|
>    | L2 Loss       | 20.97/0.752| 23.97/0.818 | 23.21/0.791 | 24.28/0.827 |
>    | SSIM Loss     | **22.39/0.793** | **25.16/0.869** | **24.55/0.800** | **25.16/0.879** |
>
>
> ---
>
> **[W2.4]** *Model Complexity Comparison:*
>
> Thanks for the suggestion. Please refer to the global response for the response.
>
>
> ---
>
> **[Q1]** *To my knowledge, the performance of fully supervised methods highly depends on the training data. So I would like to know how the supervised methods were trained on the Lai, Kohler, and Levin datasets, since only test images are contained in them:*
>
> Please refer to the global response for the details.
>
> ---
>
>
> **[Q2]** *The method is constructed based on the uniform blurring assumption, and thus it should be discussed why it can perform well on the Kohler dataset which focuses on non-uniform blurring:*
>
> The Köhler dataset is generated by taking pictures of an image (the ground truth) posted on a board. The degree of non-uniform blurring is not severe compared to the overall blurring degree. Uniform blurring can provide a reasonably effective approach to handling its blurring effect. The experiment on the Köhler dataset is commonly used to test the robustness of uniform deblurring methods on images with modest non-uniform blurring. We will clarify this in the re vision to prevent any potential confusion.

---

### Official Review · Reviewer_QZrB · 2024-07-13

**Soundness:** 3
**Presentation:** 2
**Contribution:** 3
**Rating:** 6
**Confidence:** 4

**Summary:**

This paper presents an approach to solve blind image deconvolution. The authors use multiscale Implicit Neural Representations (INRs) to depict both the latent image and the blur kernel. In addition, they propose a method that incorporates a cross-scale consistency loss and a progressive scale optimization process. Experimental results demonstrate superior performance on small-scale simulated datasets and competitive results compared to state-of-the-art methods on large-scale and real-world datasets.

**Strengths:**

1. The authors propose a cross-scale loss function that compensates for the inaccuracies introduced by simply downscaling the latent image and the blur kernel, and then calculating their convolution at the lower scale. This is achieved through the use of Quadrature Mirror Filters (QMFs).

2. The authors conduct ablation studies, providing evidence of the effectiveness of each component within the proposed method.

3. The proposed method outperforms other considered unsupervised methods in synthetic uniform blind deconvolution.

**Weaknesses:**

1. There exist prior works on Multiscale Implicit Neural Representations. The authors should provide a more detailed discussion on the similarities and differences between their proposed method and these previous works. For instance:


- “PINs: Progressive Implicit Networks for Multi-Scale Neural Representations” by Landgraf, Zoe, Alexander Sorkine Hornung, and Ricardo S. Cabral, presented at ICML 2022.

- “Miner: Multiscale Implicit Neural Representation” by Saragadam, Vishwanath, et al., presented at ECCV 2022.


2. The method appears to require at least one hyperparameter (the number of scales) to be set based on the target dataset. This suggests that the method is not entirely blind.

**Questions:**

\- Minor points:

    \- In Algorithm 1, please indicate whether the loop end condition in step 4 includes ‘0’.

    \- On line 259, the sentence “the test dataset consists of 120 images“ appears to be repeated.

**Limitations:**

Blind deconvolution, when limited to the uniform case, may not be particularly useful for real-world problems. While the authors have acknowledged this, it would be beneficial to include experiments demonstrating the effectiveness of the proposed method’s components on widely-cited non-uniform datasets, such as GoPro [40]. This would provide valuable guidance for future research in the field.

---

> ### Author Rebuttal · Authors · 2024-08-06
>
> Thank you for appreciating our work and your valuable comments. See below for the responses to the concerns and questions.
>
>
> **[W1]** *There exist prior works on Multiscale Implicit Neural Representations. The authors should provide a more detailed discussion on the similarities and differences between their proposed method and these previous works. For instance, "PINs" (Landgraf et al., ICML 2022) and "Miner" (Saragadam et al., ECCV 2022).*
>
> Thanks for the references. The concepts of "progressive" and "multi-scale" have different definitions in PINs/Miner and our work, as they have different aims and applications. PINs and Miner focus on designing **NN architectures** for INRs to achieve higher image compression ratios. In contrast, our goal is to develop new **training scheme and loss functions**, utilizing the resolution-free property of an off-the-shelf INR, for solving the blind image deconvolution problem. Specifically,
>
> +   "Progressive" in PINs refers to the progressive encoding of frequencies  in NN design. In our approach, "progressive" refers to training the NN from a coarse scale to a finer scale in the image domain, which pertains to the training process rather than NN  design.
>
> +   "Multi-scale" in PINs refers to employing a multi-scale implementation w.r.t. different frequency spectra for positional encoding and a hierarchical MLP to encode these spectra. In Miner, the image is represented in a multi-scale pyramid with correspondingly varying sizes of MLPs. Again, both approaches concern NN design. In contrast, our approach involves utilizing the cross-scale property of convolution and progressive training from coarse scale to fine scale, focusing solely on  training.
>
> We will include such a discussion in the revision.
>
> ---
>
> **[W2]** *The method appears to require at least one hyperparameter (the number of scales) to be set based on the target dataset. This suggests that the method is not entirely blind.*
>
> Following common definitions, in the context of image deblurring, "blind image deconvolution" refers to estimating both the kernel and the image from a blurred image, as opposed to "non-blind image deconvolution," which refers to estimating only the image from a blurred image and a known kernel. Thus, while "blind" in general image restoration may refer to the automated setup of all parameters, in our context, it specifically refers to the unknown nature of the blur kernel. We will clarify it  in the revision. In addition, the number of scales is dependent on the image size of the dataset, which is easy to set in practice.
>
> ---
>
> **[Q1]** *In Algorithm 1, please indicate whether the loop end condition in step 4 includes '0'.*
>
> We appreciate the reviewer's attention to detail. The loop end condition in step 4 of Algorithm 1 does  include '0'. We will make it clear in the revision.
>
> ---
>
> **[Q2]** *On line 259, the sentence "the test dataset consists of 120 images" appears to be repeated.*
>
> Thank you for pointing out the typo. We will correct it in the revision.
>
> ---
>
> **[Limitation]** *It would be beneficial to include experiments demonstrating the effectiveness of the proposed method’s components on widely-cited non-uniform datasets, such as GoPro [40]. This would provide valuable guidance for future research in the field.*
>
> Thanks for the comments. Blind deblurring has three types of configurations: uniform blurring, non-uniform blurring of static scenes, and dynamic scenes with moving objects. To the best of our knowledge, there is no self-supervised learning method for deblurring dynamic scenes yet. While our method is optimized for uniform blurring, we also evaluate its performance on both synthetic datasets with non-uniform blurring (Köhler dataset in Section 4.1) and real-world non-uniform blurring datasets of static scenes (RealBlur in Section 4.2). The results show the robustness of our approach to non-uniform blurring of static scene. GoPro, a dataset for dynamic scene deblurring, is not included for evaluation, as the image formation model used in our approach cannot model the blurring caused by moving objects. We will discuss this limitation and provide a few examples on the GoPro dataset in revision.

---

> > ### Comment · Reviewer_QZrB · 2024-08-13
> >
> > I appreciate the authors’ efforts to address the comments made by the reviewers. The responses made by the authors, the promised changes to the text, and the additional results provided seem to have addressed many of the concerns raised. I look forward to seeing the updated feedback from the other reviewers.

---

### Author Rebuttal · Authors · 2024-08-06

Dear AC and reviewers,

We sincerely appreciate the reviewers for their constructive comments, as well as their time and effort in evaluating our manuscript.
Please find below our clarifications on some common concerns and questions.

---

 **Main contributions**

Our work presents two main contributions:

+   A novel cross-scale consistency constraint for regularizing self-supervised BID.
+   A progressive coarse-to-fine training scheme to effectively alleviate overfitting of NN caused by solution ambiguities.

The main differences between the proposed one and existing works are:

+ The usage of Implicit Neural Representation (INR) for representing kernels/images is not our main contribution, our INR-related contribution is the first to exploit the resolution-free property of coordinate-NN for image restoration. This resolution-free property is crucial for the effectiveness of our proposed constraint and training scheme. Our ablation study demonstrates that direct down-sampling on image-to-image NN does not work for the proposed constraints and training scheme.

+   While there are existing works on multi-scale INR for image processing, their aims and focuses are different. For general image representation, the multi-scale approach focuses on NN design for compact representation, rather than on training or loss functions. In super-resolution (SR), the task differs significantly from BID, although both involve kernel estimation.

+    Progressive training has also been employed in traditional iterative methods, but our work represents the first implementation in a deep learning-based approach. A straightforward downsampling-based generalization to image-to-image NNs does not work; the resolution-free property of INR is crucial to its success.


Although our method is specifically designed for BID, the concept of utilizing the resolution-free property of INR has potential applications in developing  multi-scale approaches to other  image restoration tasks.

---

**Comparison to supervised methods**

We sincerely thank the reviewers for pointing this out. As this paper's focus  is on developing a powerful self-supervised method, our experiments focus on the performance gains  of our method achieves over existing self-supervised methods. Consequently, we have directly cited the results of supervised methods from related works [21, 32] for consistency. These results were obtained from the models trained on the GoPro dataset [40], which covers non-uniform blurring of dynamic scenes.

We agree that the performance of a supervised method will be impacted  by training data. In response, we have re-trained two representative supervised methods, Restormer [73] and MPRNet [74], on a dataset with only uniform blurring, the BSD-D dataset provided by "Real-blur" [1]. This dataset contains 20,000 image pairs generated by 40 synthetic uniform motion blur kernels using the method from [Schmidt et al. CVPR'16]. Please refer to Table 1 below for the results.


*Table 1: Average PSNR/SSIM of the results from compared supervised methods and Ours*
| Method     | Training data| Lai et al.’s Dataset | Köhler’s Dataset | Levin et al.’s Dataset |
|---------------|--------------------|--------------------|------------------|------------------------|
| Restormer[73]    |Non-uniform |16.31 / 0.474      | 27.61 / 0.828    | 27.38 / 0.838          |
| Restormer*[73]    |Uniform| 18.89 / 0.555      | 28.25 / 0.852    | 30.30 / 0.896          |
| MPRNet[74]    |Non-uniform |    16.15 / 0.454   |  26.32 / 0.827  | 26.21 / 0.795         |
| MPRNet*[74]    |Uniform|     18.42 / 0.531   |  27.91 / 0.848  |     28.35 / 0.850      |
| Ours    |- |**25.16 / 0.879**      | **30.69 / 0.942**    | **33.74 / 0.938**

It can be seen that a supervised method trained on a dataset with only uniform blur performs better on testing data with uniform blur, when compared to one trained on the dataset with non-uniform blurring. However, this improvement is still not enough to match the performance of our method. The main reason is that existing supervised methods aim to handle general blurring and thus they do not utilize the physics prior of image formation, specifically the convolution model for uniform blurring. Consequently, they do not perform as well as our self-supervised method, which leverages this physics prior.

---

**Comparison of Model complexity & computational cost**

We would like to first point out that all existing self-supervised BID methods train an individual NN for each sample for inference. There is no model pre-training time. Please see Appendix D for the comparison of inference time among different methods. See also Table 2 below for the comparison  of different methods when processing a  image of size $256\times 256$ with a  blur kernel size $31\times 31$. The results show that while all methods have similar running times and model sizes, MCEM and ours have noticeably smaller memory usage.

*Table 2: Model complexity comparison*
| Methods | SelfDeblur [50]  | MCEM [32] | Ours  |
|-|-|-|-|
| Running time (s) | 219.71 | 226.33| 213.02|
| #Params (k)| 3427.2| 2409.0| 2342.4|
| Memory usage (GB)| 3.61 | 1.27  | 1.82  |

We will add these results in the revision.

[1] Rim et al. "Real-World Blur Dataset for Learning and Benchmarking Deblurring Algorithms", ECCV, 2020.

---

### Decision · Program_Chairs · 2024-09-25

**Decision:**

Accept (poster)

**Comment:**

This paper receives two borderline accept and two weak accept. This paper proposes a self-supervised method for blind image deconvolution. In the rebuttal, the authors have adequately addressed most of the reviewers' major concerns.  A more detailed motivation for why INR and the progressive cross-scale can benefit self-supervised learning is needed in the revised version.